# Unified Semantic Transformer for 3D Scene Understanding

**Sebastian Koch**                                               *sebastian.koch@uni-ulm.de*
*Ulm University*
*Google**

**Johanna Wald**                                                 *johannawald@google.com*
*Google*

**Hidenobu Matsuki**                                             *hidematsu@google.com*
*Google*

**Pedro Hermosilla**                                             *phermosilla@cvl.tuwien.ac.at*
*TU Vienna*

**Timo Ropinski**                                                *timo.ropinski@uni-ulm.de*
*Ulm University*

**Federico Tombari**                                             *tombari@google.com*
*Google*
*TU Munich*

**Reviewed on OpenReview:** *https://openreview.net/forum?id=eB7oHCJzud*

## Abstract

Holistic 3D scene understanding involves capturing and parsing unstructured 3D environments. Due to the inherent complexity of the real world, existing models have predominantly been developed and limited to be task-specific. We introduce UNITE, a Unified Semantic Transformer for 3D scene understanding, a novel feed-forward neural network that unifies a diverse set of 3D dense semantic indoor tasks within a single model. Our model operates on unseen scenes trained in a fully end-to-end manner and only takes a couple seconds to infer the full 3D semantic geometry. Our approach is capable of directly predicting multiple dense semantic attributes, including 3D scene segmentation, instance embeddings, open-vocabulary features, and articulations, solely from RGB images. The method is trained using a combination of 2D distillation, heavily relying on self-supervision and leverages novel multi-view losses designed to ensure 3D view consistency. We demonstrate that UNITE achieves state-of-the-art performance on several different dense indoor semantic tasks and even outperforms task-specific models, in many cases, surpassing methods that operate on ground truth 3D geometry.

## 1 Introduction

3D scene understanding is the foundation of applications in AR/VR and robotics by enabling systems to perceive their surroundings and construct rich 3D representations that combine geometry, such as meshes, point clouds, and TSDF, with high-level semantics of object entities, class semantics, their material, state, or even affordances. While recent advances in foundation models (Radford et al., 2021; Ghiasi et al., 2022; Zhai et al., 2023; Oquab et al., 2023) and multi-modal LLMs (Liu et al., 2023a;b; Dubey et al., 2024) greatly improved the extraction of semantics from 2D images; these approaches exclusively operate on 2D inputs and therefore do not fully leverage the geometric reasoning available in 3D data.

To transfer the knowledge of strong 2D models into multi-view consistent 3D representations, three main strategies have

---

*This work was conducted during an internship at Google.

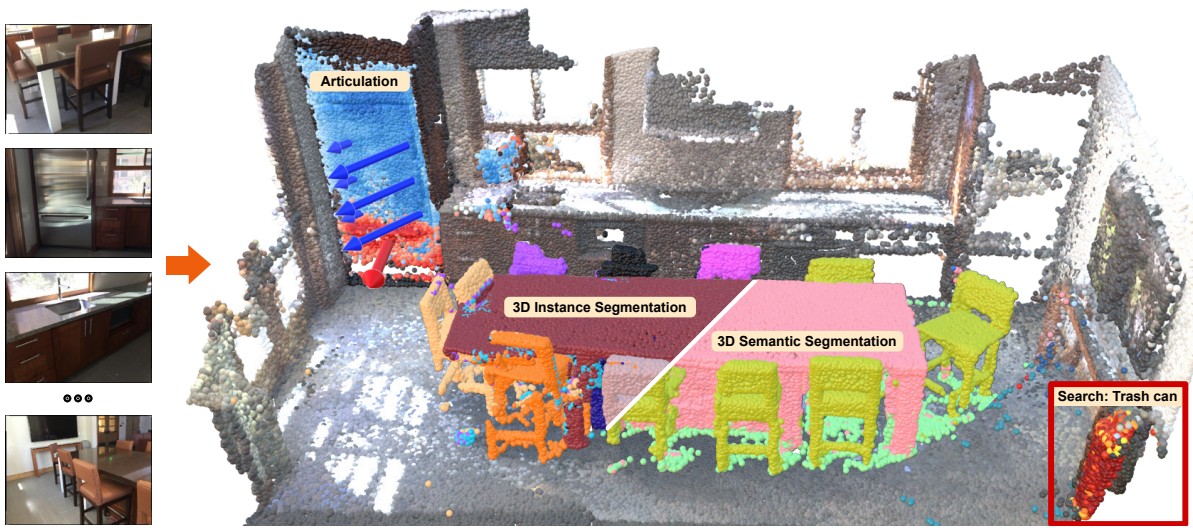

Figure 1: **UNITE Overview.** Given a set of images, UNITE reconstructs both the 3D scene geometry and 3D features used to perform semantic segmentation, instance segmentation, open-vocabulary search and articulation prediction.

emerged. First, radiance-field methods (Engelmann et al., 2024; Kerr et al., 2023; Koch et al., 2025; Qin et al., 2024) learn multi-view features from 2D foundation models alongside NeRF (Mildenhall et al., 2021) or Gaussian Splatting (Kerbl et al., 2023). They regress consistent 3D features but depend on known camera poses and scene-specific training, which does not generalize to new environments. Second, distillation methods like OpenScene (Peng et al., 2023) distill 2D features into 3D networks that operate on point clouds directly by aligning their feature spaces with respective losses. These methods then need the 3D reconstruction at inference time. Finally, some approaches propose simple but effective lifting-based techniques (Gu et al., 2024; Takmaz et al., 2025; 2023; Werby et al., 2024) to decouple 3D geometry and high-level semantics by projecting 2D predictions into an explicit 3D reconstruction. This modular design is non-differentiable and depends on hand-crafted view selection or scene segmentation, often carefully tuned for the specific application domain, which does not scale well. Since view-dependent features of 2D models are sensitive to background noise and limited context, achieving strong multi-view semantic consistency remains challenging.

Recently, transformer-based feed-forward models for 3D reconstruction have demonstrated a breakthrough in multi-view geometric consistency by unifying tasks within a single architecture. Approaches such as DUSt3R (Wang et al., 2024) and VGGT (Wang et al., 2025a) recover camera poses, depth maps, point maps, and point tracks in a single feed-forward pass from RGB images alone. Despite their success, these methods address only geometric scene attributes and do not model the semantic properties of the input. While several works have explored their extension to semantic tasks (Hu et al., 2025; Fan et al., 2024; Sun et al., 2025), they address only a limited set of tasks and rely on 2D-to-3D feature lifting. The most closely related work, IGGT (Li et al., 2025), learns instance segmentation end-to-end alongside geometry but still depends on frozen 2D foundation models at inference for open-vocabulary semantics.

This paper addresses these limitations by introducing UNITE, a large feed-forward transformer capable of unified geometry-grounded semantic prediction. Our method achieves native 3D semantic consistency by jointly learning geometry and semantics within a single network, avoiding any hand-designed lifting steps and enabling a truly end-to-end formulation for diverse semantic tasks. To this end, we propose the following contributions:

- We introduce UNITE, a semantic feed-forward transformer that predicts a full 3D reconstruction with dense 3D semantic attributes from multi-view images, including semantic and instance segmentation, open-vocabulary queries for objects, as well as object articulation.

- We train the unified model end-to-end using distillation from 2D foundation models and enforce a novel multi-view consistency loss for consistent 3D semantic predictions.

- UNITE achieves state-of-the-art results on indoor semantic tasks and outperforms zero-shot semantic feed-forward models as well as other lifting methods, even improving upon methods that operate on ground truth 3D geometry.

## 2 Related Work

**3D Scene Understanding.** Early 3D scene understanding approaches were task-specific, closed-vocabulary, and over-fitted to benchmarks, addressing specialized problems such as 3D object detection (Qi et al., 2019; Misra et al., 2021), semantic segmentation (Nekrasov et al., 2021; Wu et al., 2024), instance segmentation (Vu et al., 2022; Schult et al., 2023; Han et al., 2020) or affordance and articulation prediction (Delitzas et al., 2024; Mao et al., 2022) on point clouds and RGB-D images. These methods were typically trained on curated datasets such as ScanNet (Dai et al., 2017), 3RScan (Wald et al., 2019), or MultiScan (Mao et al., 2022), limiting their generalization beyond those domains. Recent work has shifted toward open-vocabulary prediction (Peng et al., 2023; Takmaz et al., 2023; Koch et al., 2024; Engelmann et al., 2024; Kerr et al., 2023; Qin et al., 2024), enabling more flexible deployment across diverse scenes and domains. However, this transition has been largely driven by 2D foundation models (Radford et al., 2021; Oquab et al., 2023; Kirillov et al., 2023) trained on internet-scale data, making it more challenging to develop a purely 3D open-vocabulary approach. To this end, many methods have integrated vision-language models such as CLIP (Radford et al., 2021) or SAM (Kirillov et al., 2023), not only during training (Peng et al., 2023; Koch et al., 2024; Engelmann et al., 2024) but also at inference through hand-crafted, non-end-to-end pipelines (Takmaz et al., 2023; Gu et al., 2024; Takmaz et al., 2025; Werby et al., 2024; Nguyen et al., 2024). This dependency substantially increases the computational and data requirements of 3D scene understanding pipelines, as they often rely on both complete 3D reconstructions and aligned RGB-D images with known poses and intrinsics.

**Feed-Forward Models.** Classically, image-based dense 3D reconstruction has been approached by integrating multiple subproblems, such as keypoint detection, matching, bundle adjustment, and multi-view stereo. DUSt3R (Wang et al., 2024) marked a paradigm shift, demonstrating that a single transformer could effectively solve these subproblems just by a minimal post-processing of dense point map prediction. VGGT (Wang et al., 2025a) and subsequent works (Keetha et al., 2025; Wang et al., 2025b; Liu et al., 2025) have further shown that the model can take an arbitrary number of input views and predict diverse geometric attributes.

This success has motivated efforts toward unified 3D semantic scene understanding using a similar end-to-end approach. Initial works addressed the geometric-only limitation by projecting CLIP features into 3D with DUSt3R or VGGT (Hu et al., 2025; Gong et al., 2025). Other methods introduced feature 3D Gaussian Splatting heads on top of DUSt3R and VGGT to render CLIP-aligned features (Fan et al., 2024; Sun et al., 2025). Concurrent efforts, integrate SAM features into VGGT but still rely on CLIP feature lifting for open-vocabulary segmentation (Li et al., 2025). While promising, these methods rely on semantic features lifted from 2D foundation models, resulting in a lifting process that often uses hand-designed fusion algorithms (Hu et al., 2025) and fails to achieve the native, fully learned 3D consistency seen in the original geometric models.

SAB3R (Chen et al., 2025) addressed this consistency issue to some extent by learning a dedicated semantic head directly on top of the DUSt3R backbone, yet they do not explicitly enforce any multi-view consistency between features. PanSt3R (Zust et al., 2025) achieves multi-view consistent semantic prediction through a single feed-forward pass by combining MUSt3R with a dedicated segmentation architecture, Mask2Former (Cheng et al., 2022). However, its reliance on a separate segmentation model introduces a multi-stage dependency that disentangles geometry and semantic predictions. Similarly, SIU3R (Xu et al., 2025) builds upon feed-forward models to achieve multi-view consistency via a shared query-based decoder, explicitly connecting semantic and geometry tasks through a 3D Gaussian Splatting head. Among existing feed-forward semantic approaches, IGGT (Li et al., 2025) is the most similar to ours. IGGT demonstrates that feed-forward geometric models can be extended with contrastive instance learning from SAM masks, but relies on frozen 2D open-vocabulary methods such as OpenSeg (Ghiasi et al., 2022) and does not explicitly enforce multi-view semantic consistency, ultimately diluting the gains from multi-view consistency.

In contrast, our objective is to develop a single, unified feed-forward model that jointly estimates the geometric structure and multiple dense semantic attributes for an arbitrary number of input views. Crucially, our approach does not require any hand-designed post-processing or reliance on separate task-specific networks, making it fully trainable in an end-to-end manner for holistic 3D indoor scene understanding.

## 3 Method

Given a set of RGB images $\mathcal{I} = {I_i}_{i=1}^{N}$ capturing a scene from multiple viewpoints, our goal is to reconstruct a holistic

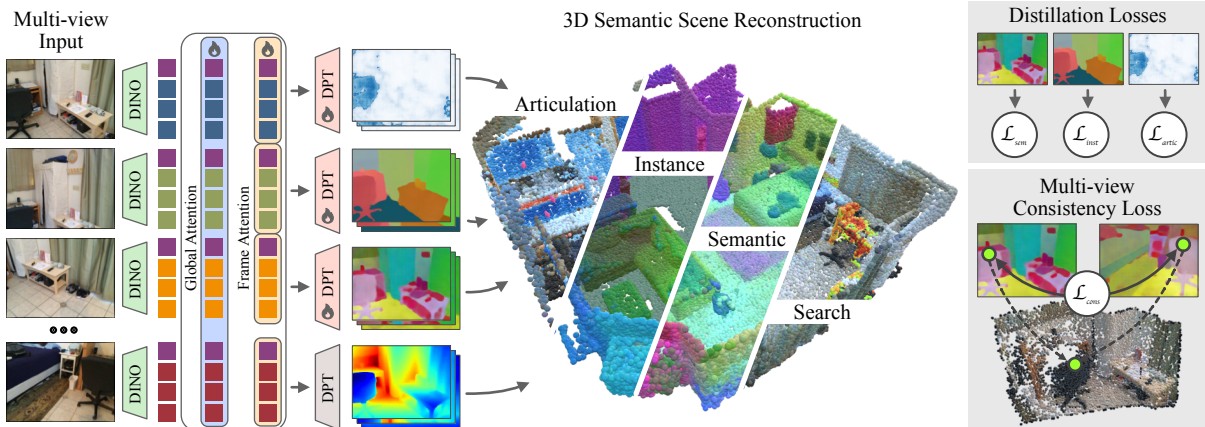

Figure 2: **Method Overview.** Given a set of RGB images, we predict a point cloud reconstruction of the scene with dense features for each point. We train a feed-forward network end-to-end to encode semantic, instance and articulation features, supervised by foundation model features and labels on 2D images. For each output view, we enforce a multi-view consistency loss for corresponding points to ensure that pixels mapping to the same point share the same feature.

3D representation that provides semantic and instance-level features at different granularities, and models object articulations alongside per-point geometry, in order to enable interactive querying of concepts and functionalities within the given scene.

To this end, we propose an end-to-end trainable multi-task network $F_\theta$ that jointly infers the 3D structure and its semantic and functional attributes in a single forward pass:

$$F_\theta\big(\{I_i\}_{i=1}^N\big) \; \mapsto \; \big\{k_i,\, D_i,\, P_i,\, S_i,\, G_i,\, A_i\big\}_{i=1}^N, \tag{1}$$

where $k_i$ denotes the camera intrinsics and pose, $D_i$ the predicted depth map, $P_i$ the point map representing per-pixel 3D coordinates, $S_i$ the semantic features (Sec. 3.2), $G_i$ the instance grouping features (Sec. 3.3), and $A_i$ the articulation predictions for each image (Sec. 3.4). All tasks are trained jointly using a combination of 2D and 3D supervision signals, complemented by a multi-view consistency objective that enforces agreement across views observing the same 3D point. An overview of the proposed framework, UNITE, is shown in Fig. 2.

## 3.1 Geometric Foundation

To extract the geometric attributes of the scene, we build upon recent pre-trained feed-forward models for geometry prediction, specifically VGGT (Wang et al., 2025a), as a foundation. A set of multi-view images is encoded using a pre-trained image encoder (Oquab et al., 2023), and the resulting image tokens are processed through $k$ blocks of alternating frame-wise and global self-attention. This backbone integrates information across views and, through dedicated heads, predicts camera poses and point maps, providing a strong geometric foundation through its large-scale pre-training and multi-view reasoning capability. The resulting representations offer a promising basis for semantic understanding, as they capture spatial consistency and geometric structure across views.

## 3.2 Semantics Features

We propose a Dense Prediction Transformer (DPT) (Ranftl et al., 2021) which learns open-vocabulary semantics on top of the shared multi-view fusion backbone. This design ensures that, in a multi-task setting, each head can learn task-specific features while sharing the strong multi-view fusion backbone. Thus, we are able to predict multi-view consistent features in CLIP space, denoted as $f_{\text{sem}} \in \mathbf{R}^{W \times H \times d_s}$, along with a per-pixel semantic confidence score $c_{\text{sem}} \in [1, \infty)$.

**Distillation.** We supervise the DPT-Head using dense 2D features $f_{\text{sem}}^{2D} \in \mathbf{R}^{W \times H \times d}$, extracted from the corresponding RGB images. To obtain these features, we first segment each image using SAM (Kirillov et al., 2023) and encode each segment using OpenSeg a pixel-aligned CLIP model, which serves as the vision-language feature extractor. The semantic distillation loss is defined as a cosine similarity loss between the predicted feed-forward features and the

CLIP-encoded 2D features:

$$\mathcal{L}_{\text{sem}}^{2D} = 1 - \cos(f_{\text{sem}}, \hat{f}_{\text{sem}}^{2D}), \tag{2}$$

which encourages the predicted features to be co-located in the vision-language representation.

**Multi-View Consistency.** We find that the 2D model provides view-inconsistent features for the same object when observed from different viewpoints, causing inconsistencies that produce a contradictory training signal, which hampers our method's goal of predicting multi-view consistent 3D semantic reconstruction (Engelmann et al., 2024; Takmaz et al., 2023). To address this issue, we introduce a multi-view consistency loss that enforces feature agreement across different projections of the same 3D point. Let $p$ denote a 3D point in the point cloud $\mathcal{Q}$, and $\mathcal{I}_p \subseteq \mathcal{I}$ the set of views from which this point is visible. Using the camera poses, intrinsics, and depth maps, we determine pixel correspondences $\text{corr}(p) = \{u_i \mid p \text{ is visible in } I_i \text{ at pixel } u_i\}$, where $u_i \in \Omega_i$ denotes the image coordinates in view $I_i$ of point $p$. For each correspondence, we compute a confidence-weighted mean feature, inspired by the success of confidence-based learning in geometric tasks (Wang et al., 2024; 2025a; Novotny et al., 2017; Wan et al., 2018)

$$\bar{f}_p = \frac{\sum_{u_i \in \text{corr}(p)} c_{p,u_i} f_{p,u_i}}{\sum_{u_i \in \text{corr}(p)} c_{p,u_i}}, \tag{3}$$

where $f_{p,u_i}$ and $c_{p,u_i}$ denote the predicted feature vector and predicted semantic confidence for point $p$ observed in view $I_i$ at pixel $u_i$, respectively. This weighted aggregation enables the model to learn which views provide more reliable features, while enforcing consistent semantics across all views of the same 3D point. For each query point $p \in \mathcal{Q}$, we first define a per-point consistency term

$$\ell_p^{\text{cons}} := \frac{1}{|\text{corr}(p)|} \sum_{u_i \in \text{corr}(p)} \left[ 1 - \cos\left(f_{p,u_i}, \text{stopgrad}(\bar{f}_p)\right) \right], \tag{4}$$

which measures the alignment between view-specific features and the aggregated feature $\bar{f}_p$, while the stop-gradient operator $\text{stopgrad}(\cdot)$ (similar to (Caron et al., 2021; Grill et al., 2020) for feature self-distillation) prevents gradients from flowing into $\bar{f}_p$. The overall multi-view consistency loss is then given by

$$\mathcal{L}_{\text{cons}} = \frac{1}{|\mathcal{Q}|} \sum_{p \in \mathcal{Q}} \ell_p^{\text{cons}}, \tag{5}$$

which encourages feature consistency across views and ensures that the learned representations remain view-invariant.

The overall semantic objective combines the distillation and consistency terms as

$$\mathcal{L}_{\text{sem}} = \lambda_{\text{sem}}^{2D} \mathcal{L}_{\text{sem}}^{2D} + \lambda_{\text{cons}} \mathcal{L}_{\text{cons}}^{\text{sem}}, \tag{6}$$

where $\lambda_{sem2D}$ and $\lambda_{cons}$ are weighting coefficients controlling the balance between 2D semantic alignment and multi-view feature consistency. This objective encourages the model to learn open-vocabulary semantic features aligned with the vision-language embedding space and consistent across multiple views of the same 3D point.

## 3.3 Instance Features

Rather than employing conventional mask-prediction networks such as Mask R-CNN (He et al., 2017) or Mask2Former (Cheng et al., 2022), which infer discrete per-view instance masks, we adopt a contrastive formulation that learns a metric embedding space. In this space, features of pixels belonging to the same instance are pulled together, while those from different instances are pushed apart. This design naturally supports multi-view consistency, as instance identities are aligned through feature similarity instead of explicit mask matching, enabling consistent supervision across viewpoints. To this end, following the same approach as the open-vocabulary semantic head, we predict instance features $g_{\text{inst}} \in \mathbb{R}^{W \times H \times d_g}$ through a DPT head on top of the shared multi-view backbone. This simple extension allows for instance, reasoning to leverage the fused multi-view context, while contributing complementary gradients that strengthen the shared encoder. Similar to the open-vocabulary head, this head is supervised by a 2D foundation model, requiring no manual annotations and scaling easily to large datasets. Specifically, we utilize SAM (Kirillov et al., 2023) to extract class-agnostic instance segmentation masks from the RGB input images.

To ensure multi-view consistent supervision, the 2D instance masks are first lifted into 3D space using ground-truth depth and camera parameters, and grouped using DBSCAN (Ester et al., 1996) to produce 3D consistent masks. These masks are then projected back into all views in which they are visible, enabling cross-view supervision with consistent instance correspondences. This 3D-aware distillation ensures that pixels corresponding to the same physical instance, even when observed from different viewpoints, are encoded with consistent instance features.

We formulate a pairwise contrastive loss in feature space to train the dense instance embeddings $g_{\text{inst}}$. Given two pixels $u_i$ and $u_j$ from views with corresponding instance assignments $l_{u_i}$ and $l_{u_j}$ provided by SAM, we define the instance contrastive loss as

$$\mathcal{L}_{\text{grouping}}^{2D} = \begin{cases} \|g_{u_i} - g_{u_j}\|_2, & l_{u_i} = l_{u_j}, \\ \text{ReLU}[m - \|g_{u_i} - g_{u_j}\|_2], & l_{u_i} \neq l_{u_j}, \end{cases} \tag{7}$$

where $g_{u_i}$ denotes the instance feature at pixel $u_i$, and $m$ is a margin that enforces a minimum separation between embeddings of different instances.

To further promote view-invariant instance representations, we apply the generalized multi-view consistency formulation introduced in Eq. (5), reusing the same confidence-weighted aggregation from Eq. (3) for the instance features $g_{\text{inst}}$. The final instance objective combines the grouping and consistency terms as

$$\mathcal{L}_{\text{inst}} = \lambda_{\text{group}} \mathcal{L}_{\text{grouping}}^{2D} + \lambda_{\text{cons}} \mathcal{L}_{\text{cons}}^{\text{inst}}, \tag{8}$$

where $\lambda_{\text{group}}$ and $\lambda_{\text{cons}}$ control the balance between intra-instance compactness and cross-view consistency. This formulation encourages the model to learn dense, class-agnostic instance features that cluster tightly within instances, remain well-separated across different instances, and stay consistent across all viewpoints.

## 3.4 Articulations

To extend scene understanding beyond static geometry and semantics, we additionally predict object articulations in a dense, per-pixel manner, also by using a dedicated DPT head. Articulations capture how object parts can move and provide functional information essential for reasoning about interactions within a scene.

We consider two fundamental articulation types, that we learn from annotated data in 3D: translational motions, such as drawers sliding along a linear path, and rotational motions, such as doors rotating around a hinge. To represent both within a single regression space and to ensure a dense prediction setting, we approximate rotational motion as linear ones. This approximation is computed from ground-truth articulation parameters: for each object with a known rotation axis, we rotate its surface points by $90 \deg$ around this axis and use the displacement between original and rotated positions as the linearized ground-truth motion direction.

Because articulated objects are sparse in typical scenes, we decompose the prediction into two tasks: (i) identifying which points belong to articulated objects, and (ii) predicting their local motion directions. The articulation head outputs for translation and rotation articulation separately a per-pixel articulation probability map $p_{\text{pred}} \in [0, 1]^{W \times H}$ and a per-pixel 3D *vector map* $\mathbf{v}_{\text{pred}} \in \mathbb{R}^{W \times H \times 3}$ together with a overall corresponding confidence map $c_{\text{motion}}$. These vector maps encode the articulation motion at each pixel, analogous to how point maps (Wang et al., 2024) assign a 3D coordinate to each pixel.

Articulation existence is trained using a binary ground-truth map $\hat{p}$ with a focal loss to address class imbalance:

$$\mathcal{L}_{\text{exist}} = \frac{1}{M} \sum_{i=1}^{M} \mathcal{L}_{\text{Focal}}(p_i, \hat{p}_i). \tag{9}$$

For pixels corresponding to articulated regions, motion vectors $\mathbf{v}_i$ are regressed toward ground-truth vectors $\hat{\mathbf{v}}_i$ using an $\ell_2$ loss:

$$\mathcal{L}_{\text{motion}} = \frac{1}{M_A} \sum_{i \in M_A} \|\mathbf{v}_i - \hat{\mathbf{v}}_i\|_2^2, \tag{10}$$

where $M_A$ is the set of articulated pixels.

To enforce correspondence across viewpoints, we apply the confidence-weighted multi-view consistency from Eqs. (3) and (5) to both the existence and vector-map. The full articulation loss thus becomes:

$$\mathcal{L}_{\text{artic}} = \lambda_{\text{exist}} \, \mathcal{L}_{\text{exist}} + \lambda_{\text{cons}}^{\text{exist}} \, \mathcal{L}_{\text{cons}}^{\text{exist}} \\ + \lambda_{\text{motion}} \, \mathcal{L}_{\text{motion}} + \lambda_{\text{cons}}^{\text{motion}} \, \mathcal{L}_{\text{cons}}^{\text{motion}}. \tag{11}$$

## 3.5 Multi-Task Optimization

Finally, the network is optimized end-to-end using a weighted combination of all task-specific objectives, enabling shared features to benefit from complementary supervision across tasks

$$\mathcal{L} = \lambda_{\text{sem}} \mathcal{L}_{\text{sem}} + \lambda_{\text{inst}} \mathcal{L}_{\text{inst}} + \lambda_{\text{artic}} \mathcal{L}_{\text{artic}}.[1] \tag{12}$$

# 4 Experiments

In this section, we evaluate our approach across four open-vocabulary 3D scene understanding tasks — semantic segmentation, instance segmentation, articulation prediction and open-vocabulary search — on ScanNet (Dai et al., 2017), ScanNet200 (Rozenberszki et al., 2022), ScanNet++ (Yeshwanth et al., 2023), MultiScan (Mao et al., 2022) LERF (Kerr et al., 2023). Our method achieves state-of-the-art performance on all tasks, consistently surpassing task-specific baselines and demonstrating the benefits of a unified multi-task network.

Unlike prior work that reports view-level or 2D metrics, evaluations in this paper are performed directly in 3D to assess genuine 3D scene understanding in semantic feed-forward models. Sec. 4.1 presents quantitative results for open-vocabulary 3D semantic segmentation, followed by class-agnostic instance segmentation in Sec. 4.2. In Sec. 4.3, we show the versatility of our model for articulation and affordance prediction. Finally, in Sec. 4.4, we demonstrate the open-vocabulary understanding of our model. Sec. 4.5 provides qualitative results demonstrating generalization to diverse indoor environments, and Sec. 4.6 analyzes the impact of multi-view consistency, feature aggregation, and the joint geometric-semantic formulation.[2] As our approach is largely trained self-supervised, we only compare against approaches that are trained in a self-supervised or unsupervised manner, except for articulation prediction, where we include supervised baselines to align with our assumptions.

## 4.1 3D Semantic Segmentation

**Setup.** We evaluate our method on *ScanNet* (Dai et al., 2017), *ScanNet200* (Rozenberszki et al., 2022), and *ScanNet++* (Yeshwanth et al., 2023) for 3D semantic segmentation. These datasets differ in the number and granularity of semantic classes, and we evaluate performance on all of them using standard class-wise mIoU and mAcc on the ground truth 3D point clouds.

We compare our approach against recent SLAM-based open-vocabulary methods (Gu et al., 2024; Werby et al., 2024), which jointly reconstruct the scene and extract open-vocabulary features, operating under privileged conditions with access to ground-truth depth, intrinsics, and camera poses. We also include recent semantic feed-forward approaches (Hu et al., 2025; Fan et al., 2024; Sun et al., 2025; Zust et al., 2025) using their public implementations. For fairness, all methods receive up to 200 images per scene. To isolate semantic performance and ensure comparability with SLAM-based methods, we additionally provide feed-forward models with ground-truth camera parameters and poses; however, feed-forward approaches including UNITE can predict 3D scene semantics without ground truth inputs, which are provided solely to establish a fair baseline against priviledged methods. Their image-based frustum predictions are aligned with the ground-truth point cloud using a one-nearest-neighbor mapping. If a feed-forward model cannot process all 200 images at once (Fan et al., 2024; Hu et al., 2025), we split the sequence into smaller subsets and align each subset's predictions using the corresponding ground-truth poses.

Finally, we compare against established point cloud-based open-vocabulary methods, which are the most privileged, as they operate on posed RGB-D images and have access to the evaluation meshes at inference time. We omit comparisons

---

[1]A full overview of the loss weights is given in the supplementary.
[2]Further evaluations on in-the-wild, out-of-distribution and geometry-semantic synergies can be found in the appendix.

Table 1: **3D Semantic Segmentation.** We compare our method with open-vocabulary 3D point cloud methods, SLAM approaches and semantic feed-forward models evaluated with the ground truth 3D segmentation. † For fair comparison, we evaluate Panst3R without their introduced QUBO post-processing. 3D point cloud methods and RGB-D methods assume ground truth depth and poses, likewise Pe3R and LSM also require ground truth poses to aggregate pair-view predictions to scene-level inference, for fairness all feed-forward methods also receive ground truth depth and poses.

| | *ScanNet* | | *ScanNet200* | | *ScanNet++* | |
|---|---|---|---|---|---|---|
| | **mIoU** | **mAcc** | **mIoU** | **mAcc** | **mIoU** | **mAcc** |
| *3D point cloud methods* | | | | | | |
| CLIP-FO3D  (Zhang et al., 2023) | 30.2 | 49.1 | - | - | - | - |
| OpenScene 2D  (Peng et al., 2023; Ghiasi et al., 2022) | 41.4 | 63.6 | - | - | - | - |
| OpenScene 3D  (Peng et al., 2023) | **46.0** | **66.3** | **07.3** | **14.5** | - | - |
| *RGB-D methods* | | | | | | |
| Concept-Graphs (Gu et al., 2024) | 17.1 | 29.1 | 06.0 | 11.7 | - | - |
| HOV-SG (Werby et al., 2024) | **34.4** | **51.1** | **11.2** | **18.7** | - | - |
| *RGB methods* | | | | | | |
| Pe3R (Hu et al., 2025) | 10.7 | 19.8 | 02.5 | 06.5 | 08.3 | 16.0 |
| Uni3R (Sun et al., 2025) | 29.3 | 39.4 | 04.1 | 06.8 | 05.2 | 10.8 |
| LSM (Fan et al., 2024) | 32.2 | 41.1 | 06.3 | 09.7 | 14.3 | 26.5 |
| PanSt3R† (Zust et al., 2025) | 42.6 | 49.7 | 13.3 | 19.9 | **21.6** | 31.4 |
| IGGT (Li et al., 2025) | 39.7 | - | - | - | 20.8 | - |
| **Ours** | **48.7** | **68.3** | **14.5** | **26.3** | 17.2 | **37.0** |

with open-vocabulary radiance field approaches (Kerr et al., 2023; Engelmann et al., 2024; Qin et al., 2024) since they require scene-specific training, unlike our method and the baselines.

**Class prediction.** The 3D semantic segmentation is obtained by querying the 3D semantic feature representation with the benchmark set of text labels and assigning the class with the highest similarity score to each 3D point. Querying is performed by computing the cosine similarity between the point cloud features and the text embeddings of the vision language model, specifically here the CLIP-text encoder.

**Results.** Tab. 1 compares UNITE against our point cloud-based, SLAM-based and feed-forward model-based baselines. UNITE outperforms all SLAM and semantic forward-model baselines by a large margin for both mIoU and mAcc. Our closest competitor PanSt3R (Zust et al., 2025), performs well on ScanNet++ (Yeshwanth et al., 2023) whose labels were contained in the Mask2Former (Cheng et al., 2022) training of PanSt3R (Zust et al., 2025), however on ScanNet20 and ScanNet200 our approach outperforms PanSt3R by a large margin. Furthermore, our approach significantly outperforms IGGT on ScanNet. Unlike IGGT, which struggles with multi-view inconsistencies by lifting OpenSeg (Ghiasi et al., 2022) features via VGGT (Wang et al., 2025a) geometry, we directly predict semantics with multi-view consistent features. Furthermore, our approach is the only RGB(-D) approach that outperforms a native point cloud-based 3D scene understanding approach in OpenScene (Peng et al., 2023).

## 4.2   3D Instance Segmentation

**Setup.** Instance segmentation is evaluated on *ScanNet* (Dai et al., 2017), *ScanNet200* (Rozenberszki et al., 2022), and *ScanNet++* (Yeshwanth et al., 2023), comparing our approach to representative unsupervised, lifting-based, and feed-forward methods. We omit comparisons with approaches trained on instance mask annotations, such as Mask3D (Schult et al., 2023), since these methods rely on dense supervision unavailable in our setting, making such comparisons unfair and not representative of the unsupervised or weakly supervised nature of our approach. Following the ScanNet evaluation protocol, we report Average Precision (AP) at mask overlap thresholds of 50% and 25%, as well as the mean AP averaged over IoU thresholds from 0.50 to 0.95 in steps of 0.05. Because our model performs class-agnostic instance segmentation, we ignore semantic class labels when matching predictions to the ground-truth instances.

We compare against several class-agnostic baselines: HDBSCAN (McInnes & Healy, 2017), which clusters instances purely from geometry; Felzenszwalb et al. (Felzenszwalb & Huttenlocher, 2004), which applies a graph-based segmentation approach; SAM3D (Yang et al., 2023), which lifts 2D SAM (Kirillov et al., 2023) masks into 3D using ground-truth depth and camera parameters and PanSt3R (Zust et al., 2025), which employs a Mask2Former (Cheng et al., 2022) head for instance prediction;.

**Instance Feature Clustering.** Our method produces dense instance features, which we cluster to obtain the final class-agnostic 3D instance segmentation. When the number of instances is unknown, we apply HDBSCAN

Table 2: **Class-agnostic 3D Instance Segmentation.** We compare our method against different 3D instance segmentation approaches † For fair comparison, we evaluate Panst3R without their introduced QUBO post-processing. Please note, that unlike the other approaches, PanSt3R requires instances labels for training. HDBSCAN and Felzenszwald operate on GT 3D point cloudss, while SAM3D utilizes depth, intrinsic and GT poses. For fairness we provide the feed-forward models with GT depth and poses.

| | ScanNet | | | ScanNet200 | | | ScanNet++ | | |
| --- | --- | --- | --- | --- | --- | --- | --- | --- | --- |
| | AP | $AP_{50}$ | $AP_{25}$ | AP | $AP_{50}$ | $AP_{25}$ | AP | $AP_{50}$ | $AP_{25}$ |
| HDBSCAN (McInnes & Healy, 2017) | 1.6 | 5.5 | 32.1 | 2.9 | 08.2 | 33.1 | 4.3 | 10.6 | 32.3 |
| Felzenszwalb (Felzenszwalb & Huttenlocher, 2004) | 5.3 | 12.6 | 36.9 | 04.8 | 09.8 | 27.5 | **08.8** | **16.9** | 36.1 |
| SAM3D (Yang et al., 2023) | 6.3 | 17.9 | 47.3 | 12.1 | 28.6 | 54.1 | 03.0 | 7.9 | 22.3 |
| PanSt3R † (Zust et al., 2025) | 11.4 | 29.3 | 53.4 | 10.6 | 27.3 | 50.8 | 06.5 | 15.9 | 33.9 |
| IGGT (Li et al., 2025) | 12.3 | 24.9 | 41.6 | - | - | - | - | - | - |
| **Ours** | **13.2** | **29.6** | **57.2** | **12.3** | 28.8 | **58.2** | 06.6 | 16.1 | **40.4** |

Table 3: **3D Object Articulation Prediction.** We compare against the point cloud-based articulation methods on MultiScan.

| | IoU | Movable Part | | | Motion type | | |
| --- | --- | --- | --- | --- | --- | --- | --- |
| | | R | P | F1 | R | P | F1 |
| OPDPN (Jiang et al., 2022) | 53.8 | 01.6 | 03.1 | 02.1 | 01.3 | 02.6 | 01.7 |
| S2M (Wang et al., 2019) | 69.2 | **06.7** | 15.7 | 09.4 | **04.5** | 10.4 | 06.3 |
| Ours (task-only) | 68.3 | 05.2 | 23.4 | 07.6 | 01.8 | 09.2 | 04.9 |
| Ours (multi-task) | **70.3** | 06.0 | **29.6** | **10.0** | 03.7 | **12.3** | **06.9** |

(McInnes & Healy, 2017). When the number of instances is known or predefined, we instead use KMeans++ (Hartigan & Wong, 1979).

**Results.** Tab. 2 reports our class-agnostic 3D instance segmentation results. UNITE achieves state-of-the-art performance on ScanNet and ScanNet200, surpassing other unsupervised approaches, including SAM3D (Yang et al., 2023), which lifts SAM predictions into 3D using depth maps and merges them via IoU heuristics. Although, UNITE is also trained with SAM masks, our multi-view consistency losses enable significantly stronger 3D-consistent predictions. Moreover, UNITE outperforms the feed-forward baseline PanSt3R (Zust et al., 2025), the only method trained directly on instance labels. On ScanNet++, however, the classical method of Felzenszwalb et al. (Felzenszwalb & Huttenlocher, 2004) performs best, even surpassing our approach on AP and $AP_{50}$. We attribute this to (i) their direct operation on the evaluation mesh, avoiding lifting errors, and (ii) their advantage on fine-grained boundaries, while UNITE yields better $AP_{25}$, reflecting stronger instance-level recognition.

## 4.3 Articulation

Finally, we evaluate our method on affordance detection and articulation prediction on the MultiScan (Mao et al., 2022) dataset. The MultiScan dataset is an extensive dataset that consists of multiple rescans of the same scene to identify articulated objects. Unlike benchmarks such as SceneFun3D (Delitzas et al., 2024), which mainly focus on small, affordable elements such as buttons, knobs, and handles, MultiScan focuses on a more holistic understanding of articulated objects with a greater variety, such as drawers, cabinets, rotatable chairs, curtains, etc. This makes MultiScan ideal to evaluate our semantic feed-forward model to test the reconstruction quality and semantic understanding.

**Setup.** We evaluate UNITE against strong baselines (Jiang et al., 2022; Wang et al., 2019) from the MultiScan benchmark. Unlike our approach, these methods operate on reconstructed meshes and predict articulated object parts and their motion type (translation or rotation) in two stages: instance segmentation followed by part segmentation. In contrast, UNITE uses only input images and directly predicts articulated object parts in 3D.

We report IoU, measuring the overlap between our articulation existence prediction and ground-truth articulated parts. In addition, we compute precision, recall, and F1 for movable part detection, considering a part correctly detected if its IoU with the ground truth exceeds 0.5. Finally, we evaluate motion type prediction (translation vs. rotation) under the same IoU threshold to assess the correctness of the predicted articulation type.

**Part Articulation Aggregation.** UNITE outputs per-pixel articulation vectors together with an articulation-existence probability. To obtain a single articulation prediction for each object or object part, we first discard all motion vectors whose existence probability is below 0.5. The remaining vectors for each predicted instance are then averaged. This yields one unified articulation estimate for every instance segment as visualized in Fig. 3c.

Table 4: **LERF Scenes Evaluation.** We evaluate UNITE on LERF scenes for open-vocabulary search. The LERF dataset contains 4 scenes, with each scene having five search queries. The evaluation metric is mIoU segmentation.

| | *ramen* | *figurines* | *teatime* | *kitchen* | Avg. |
|---|---|---|---|---|---|
| LERF (Kerr et al., 2023) | 28.2 | 38.6 | 45.0 | 37.9 | 37.4 |
| Pe3R (Hu et al., 2025) | **30.5** | 40.9 | 55.9 | 38.5 | 41.4 |
| PanSt3R (Zust et al., 2025) | 18.2 | 37.9 | 61.1 | **39.0** | 39.1 |
| **Ours** | 21.9 | **43.2** | **62.8** | **39.0** | **41.7** |

Table 5: **Ablations.** We compare a naïve CLIP+VGGT baseline, against our unified geometry–semantics model on ScanNet20. We incrementally add multi-view consistency loss, semantic confidence weighting, and ground truth geometry.

| | mIoU | mAcc |
|---|---|---|
| VGGT + OpenSeg Lifting (2D Teacher) | 34.5 | 46.1 |
| **Ours** (distillation-only) | 44.3 | 61.9 |
| + Multi-View consistency (Eq. (5)) | 45.2 (+0.9) | 63.2 (+1.3) |
| + Confidence Weight (Eq. (3)) | 46.3 (+1.1) | 64.7 (+1.5) |
| + Geometry Oracle | **48.7** (+2.5) | **68.3** (+3.7) |

**Results.** Overall, UNITE outperforms the 3D point cloud baselines on articulation prediction, as shown in Tab. 3. While existing methods rely on reconstructed meshes and explicit part segmentation, our image-based approach achieves higher IoU and consistently better F1 scores for both movable part and motion type prediction. Notably, our multi-task design, which jointly learns class and instance semantics together with articulation, leads to substantial gains over the task-specific variant. However, while our precision and F1 scores are consistently higher on all metrics UNITE's recall is consistently lower than the S2M (Wang et al., 2019) baseline. We attribute this to the inherent precision-recall trade-off and our introduction of a focal loss in Eq. (9). This highlights the benefit of integrating related scene understanding objectives within a single end-to-end feed-forward network, which underlines the benefits of unified, multi-task 3D scene understanding.

## 4.4 Open-Vocabulary Search

**Setup.** Unlike semantic segmentation benchmarks where the class labels are selected from a predefined set which models frequency of classes, in open-vocabulary benchmarks the query classes are purposely chosen to be rare long-tail distribution classes. We evaluate UNITE for open-vocabulary understanding on the LERF (Kerr et al., 2023) benchmark, a benchmark consisting of 4 scenes with 5 open-vocabulary queries each such as "copper-bottom pot".

**Querying.** The localization and segmentation of the open-vocabulary queries follows the querying process described in Sec. 4.1 by computing the softmax-normalized cosine similarity between the predicted features and the text embeddings.

**Results.** Tab. 4 shows that UNITE outperforms other open-vocabulary baselines on challenging open-vocabulary queries. UNITE outperforms all baselines overall, only Pe3R (Hu et al., 2025) reports better numbers on the *ramen* scene, while UNITE produces the best segmentation performance on all other scenes.

## 4.5 Qualitative Results

In Fig. 3, we show predictions of our approach for 3D semantic and instance segmentation, open-vocabulary instance search, and articulation prediction across scenes from ScanNet (Dai et al., 2017), ScanNet++ (Yeshwanth et al., 2023), and MultiScan (Mao et al., 2022). In Fig. 3a, we perform class-agnostic 3D instance segmentation by clustering the predicted instance features (right) using HDBSCAN (McInnes & Healy, 2017). We then assign semantic labels from the ScanNet200 label set by computing cosine similarity between the predicted semantic embeddings and the encoded text label embeddings. The resulting segments exhibit clean object boundaries with minimal noise and accurate semantic predictions, demonstrating geometrically and semantically coherent 3D outputs.

In Fig. 3b, we conduct open-vocabulary instance search using CLIP-aligned features to localize both concrete objects (e.g., a billiard table, bean bags) and higher-level concepts such as sleeping, music, and exercise. In Fig. 3c, we predict object articulations in a laundry room, correctly identifying movable cabinets and the dryer door along with their estimated motion directions.

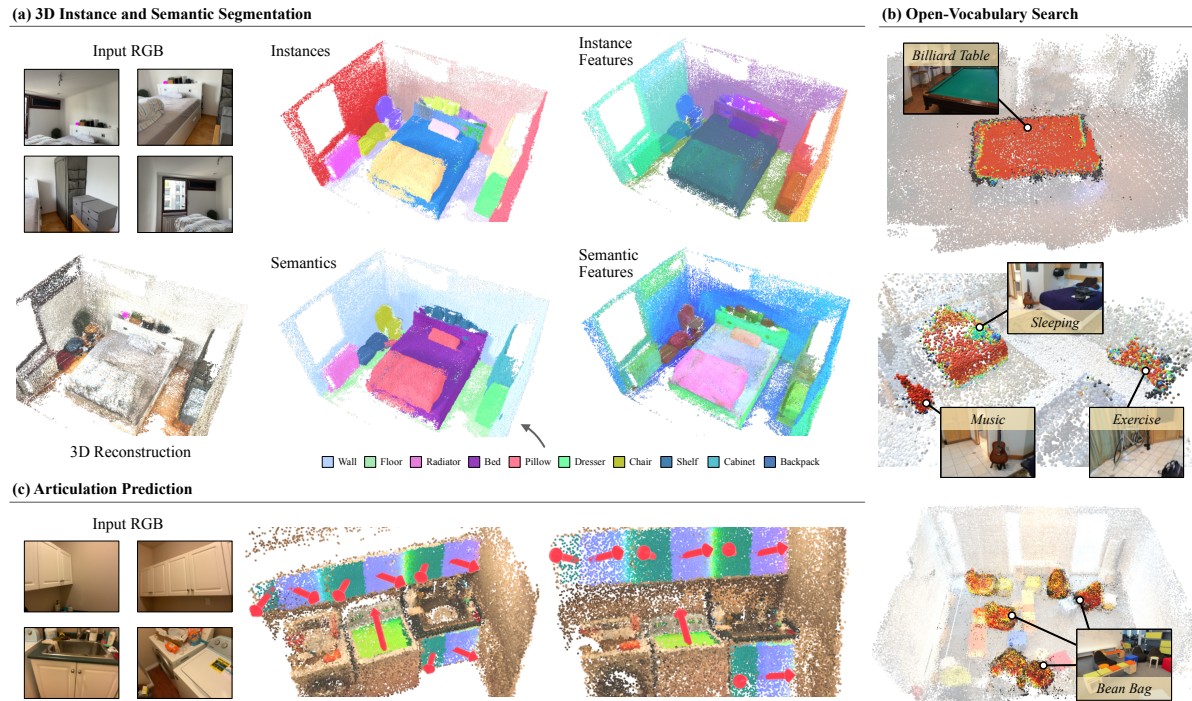

Figure 3: **Results of UNITE on 5 scenes in ScanNet, ScanNet++, and MultiScan.** Each row shows qualitative results a different output modality: (a) 3D semantic and instance segmentation, (b) open-vocabulary search and (c) articulation prediction.

## 4.6 Ablations

We perform ablation studies on four components of our model: joint training versus independent geometric and semantic models, the effect of our multi-view consistency loss with confidence-weighted fusion, the advantage of a multi-task approach over a single task approach, and an oracle-geometry upper bound using ground-truth depth and pose. For comprehensive ablations, we investigate the following questions mainly on the tasks of semantic segmentation (which does not require any post-processing, making results clear and fair) and articulation prediction:

**How does distillation compare to feature reprojection?** In this paper, we propose to distill vision-language features from CLIP (Radford et al., 2021) features into a feed-forward model that jointly reconstructs the scene and predicts its semantics. Alternatively, one could combine a feed-forward model such as VGGT (Wang et al., 2025a) with a semantic foundation model like CLIP by lifting CLIP predictions into 3D using the depth and camera parameters predicted by VGGT which essentially represents our teacher features. However, as shown in Tab. 5, this simple lifting approach performs notably worse than our end-to-end model, which natively performs multi-view fusion. Our approach produces more robust predictions by jointly reasoning across views, whereas the single-view lifting approach is limited by viewpoint occlusions and can only aggregate information through mean pooling without comprehensive multi-view consistency.

**Does semantic confidence improve multi-view fusion?** In Eq. (3), we introduce a method to learn multi-view consistent features via a confidence-weighted average of all predicted view features. The per-view confidence encourages the model to rely more on views with clear visibility, while features from occluded or uncertain views are pulled toward those from more confident perspectives. As shown in Tab. 5, this weighted averaging leads to a significant improvement over standard mean pooling. While mean pooling produces a suboptimal embedding that treats all views equally, our confidence-weighted fusion yields a more stable training objective and better overall results.

**Does multi-task training show synergy effects between tasks?** In Eq. (12), we define the loss to train our network with a multi-task objective for a unified architecture. We argue that training a unified model on multiple tasks enables the network to leverage synergies across task-specific representations, resulting in improved performance on each individual task. In Tab. 3, we show that UNITE trained in a multi-task setting outperforms UNITE trained only on the articulation task. Concretely, the *multi-task* model improves articulation prediction by +2 mIoU and motion-type F1 by

+2% compared to *task-only* (articulation-only) training.

**What if geometry were perfect?** When evaluating 3D semantic or instance segmentation, the quality of the geometric reconstruction directly affects the semantic matching, as correspondences are established via nearest-neighbor search between predicted and ground-truth points. To disentangle these factors, we report an upper bound of our method in Tab. 5, obtained by using ground-truth depth and camera parameters. The results show a modest but consistent improvement, indicating that our approach already produces geometry of sufficiently high quality for reliable semantic reasoning.

## 5 Conclusion

In summary, we introduced UNITE, a unified transformer architecture that achieves 3D semantic scene understanding directly from RGB images by distilling rich 2D foundation model features and enforcing multi-view consistency. While state-of-the-art on multiple tasks, the model's performance is currently tightly coupled to the quality of the underlying geometry. However, rapid advancements in geometric foundation models are directly transferable, with promising immediate improvements of the semantic features.

Also, while we showed the generalizability of the unified model, we only scratched the surface of what we believe is possible. Given the scalability of our training pipeline, the model has the potential of being adapted to larger datasets and expanding to more tasks like scene captioning and decomposition or 4D dynamic segmentation. We believe this opens opportunities for in-the-wild, scalable 3D scene analysis in a truly holistic and unified manner.

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

## A  Additional Details

**Implementation Details.** Our model is built on top of VGGT-1B (Wang et al., 2025a) and is trained on sequences of 12 images. To obtain semantic features, we employ SAM (Kirillov et al., 2023) to generate class-agnostic segments for each image. We then extract vision–language features by running OpenSeg (Ghiasi et al., 2022) over the full image and averaging the resulting features within each SAM segment. This produces a dense feature map of shape $H \times W \times d_s$ with $H = 378, W = 448$ and $d_s = 718$.

For instance feature supervision, we again rely on SAM to extract 2D segments. These segments are lifted into 3D using ground-truth camera parameters and depth maps. We compute 3D IoU between lifted segments and merge them following the procedure of SAM3D (Yang et al., 2023). The merged 3D segments are then projected back to the image plane, yielding multi-view consistent mask identifiers for each image. During training, the contrastive margin is set to $m = 1$, and the instance embeddings have dimensionality $d_g = 512$ to ensure sufficient separability in large-scale scenes. At inference time, instance features are clustered using HDBSCAN with eps $= 0.1$.

Articulations are represented using a 9-dimensional vector that includes three translational and three rotational components, a translation-existence mask, a rotation-existence mask, and a prediction confidence value.

The final multi-task objective assigns equal weight to the three main tasks, with $\lambda_{\text{sem}} = 1$, $\lambda_{\text{inst}} = 1$, and $\lambda_{\text{artic}} = 1$. For each task, we apply a 10-to-1 weighting ratio between the distillation/supervision loss and the multi-view consistency loss. This yields $\lambda_{\text{sem}}^{2D} = 1$ and $\lambda_{\text{cons}} = 0.1$ for semantic feature learning; $\lambda_{\text{group}} = 1$ and $\lambda_{\text{cons}} = 0.1$ for instance feature learning; and $\lambda_{\text{exist}} = 10$, $\lambda_{\text{cons}}^{\text{exist}} = 1$, $\lambda_{\text{motion}} = 1$, and $\lambda_{\text{cons}}^{\text{motion}} = 0.1$ for articulation learning. The model is trained for $150,000$ steps with a learning rate of $3e-4$ and exponential learning rate decay.

**Evaluation Details.** For each scene, we evaluate the predictions from 200 frames. The predicted point maps are first projected into 3D, at a resolution of $378 \times 448$ this generates $\sim 34M$ points. We apply farthest-point sampling to downsample the resulting point cloud to $200K$ points for efficiency. We then aggregate the embeddings of the sampled points with those of the rejected points in their respective local neighborhoods by computing a greedy matching using minimum distances. Finally, we compute 1-nearest-neighbor correspondences (see scikit NearestNeighbors) to the ground truth point cloud as a many-to-one mapping to align our prediction with the ground truth point cloud labels for the evaluation metrics. For evaluation there no point cloud preprocessing involved, the evaluation is performed on the raw benchmark point clouds. For a fair comparison with point cloud based approaches and privileged SLAM approaches, we provide all forward models with ground truth depth and poses in the evaluation of Tab. 1 and Tab. 2. Please note however this is only required for fair evaluation purposes, at inference time UNITE can be run entirely from images alone.

To align the predicted point cloud with the ground-truth coordinate system, we fix the reference frame using the ground-truth camera pose of the first image, consistent with VGGT's convention of treating the first camera as the reference. We estimate the global scale factor for alignment by solving a least-squares problem over the predicted and ground-truth camera poses.

## B  Compute and Inference Time

As detailed in Tab. 6, we report inference runtime and peak GPU memory for the full forward pass, including the semantic, instance, and articulation heads, across varying numbers of input frames. All measurements are obtained on a single NVIDIA H100 GPU using a JAX implementation with flash attention. Input images are rendered at a resolution

Table 6: **Runtime and peak GPU memory usage across different numbers of input frames.** Runtime is measured in seconds, and GPU memory usage is reported in gigabytes. The input resolution is $378 \times 448$.

| Input Frames | 2 | 10 | 20 | 100 | 200 |
|---|---|---|---|---|---|
| Time | ~5s | ~7s | ~9s | ~23s | ~34s |
| Mem. | ~2GB | ~4GB | ~14GB | ~21GB | ~40GB |

Table 7: **Segmentation performance across different numbers of input views.** mIoU and mAcc are reported in percent.

| Input Frames | 50 | 100 | 150 | 200 |
|---|---|---|---|---|
| mIoU | 29.9 | 38.5 | 47.4 | 48.7 |
| mAcc | 37.6 | 52.1 | 67.6 | 68.3 |

of $378 \times 448$. To balance inference speed and memory consumption, the DPT heads are executed sequentially for each output image, which reduces peak memory usage.

## C    Scene-scale frame scaling

A core design goal of UNITE is to scale to scene-scale inference, processing up to 200 frames per scene as demonstrated in the experiments of the main paper. To highlight the importance of being able to scale to scene-level frame counts, we provide an ablation on ScanNet (Dai et al., 2017) that varies the number of input views across $50/100/150/200$, sampling frames equidistantly per scene. As shown in Tab. 7, there is a large performance gap between 50 and 200 views, since scene coverage is severely limited with fewer frames and large-scale indoor scenes cannot be adequately captured. In contrast, the difference between 150 and 200 frames is small, indicating that once view coverage of the scene is sufficient, additional frames alone do not improve results much further. This confirms that scaling to scene-level frame counts is essential for scene understanding, and that reducing the number of frames directly degrades performance.

## D    Geometry and Semantic Synergy

The aim of our work is to bring semantic information into models that mainly rely on geometric signals. For this purpose we are using the pre-trained geometric features of VGGT (Wang et al., 2025a) as a strong starting point for learning semantics. We also expect that learning semantic features can, in turn, support geometric tasks. To test this idea, we run a small ablation on depth estimation and compare VGGT's depth predictions in three setups: the original pre-trained model, fine-tuning on ScanNet (Dai et al., 2017) with only geometric losses for $10,000$ steps, and fine-tuning on ScanNet with both geometric and semantic losses for $10,000$ steps.

As shown in Tab. 8, fine-tuning with geometric losses brings only a slight gain over the pre-trained model. This is reasonable since the model is already strong in geometric reasoning and was already trained on ScanNet. Notably, introducing our proposed semantic losses adds small improvements also in depth prediction, indicating that jointly learning geometric and semantic representations could also benefit geometric learning. However, the margins are still relatively small but we believe these results could also transfer and scale when trained on larger datasets.

## E    Out-of-Distribution Evaluation

**Replica.** In Tab. 9, we report 3D semantic segmentation performance on the Replica dataset (Straub et al., 2019) following the same evaluation protocol as for ScanNet (Dai et al., 2017) and ScanNet++ (Yeshwanth et al., 2023). This experiment highlights the out-of-distribution generalization capability of our method, which surpasses existing open-vocabulary baselines. In Fig. 4, we provide additional, qualitative results.

**LERF Scenes.** In Fig. 5, we present predictions on the kitchen scene from the *in-the-wild* LERF dataset. UNITE adapts robustly to such unconstrained settings, accurately grounding complex textual queries such as *cabinet above the refrigerator* or *knives hanging on the wall*. It further distinguishes fine-grained spatial relationships, for instance, correctly differentiating the cabinets beneath the kitchen counter and those above the sink, from the cabinet above the refrigerator.

Table 8: **Depth fine-tuning ablation.** We evaluate depth prediction after fine-tuning VGGT (Wang et al., 2025a) on ScanNet using geometric and semantic losses.

| | AbsRel↓ | $\delta_{1.25}$↑ |
|---|---|---|
| VGGT (pre-trained) | 0.260 | 63.31 |
| VGGT (fine-tuned geometry) | 0.256 | 63.52 |
| VGGT (fine-tuned geometry & semantics) | **0.251** | **63.77** |

# F   Comparison with concurrent work (IGGT)

IGGT is concurrent work that focuses on instance segmentation, learning contrastive instance features from a VGGT (Wang et al., 2025a), very similar to our approach in Sec. 3.3. Additionally, IGGT combines their instance predictions with frozen OpenSeg (Ghiasi et al., 2022) features extracted separately on the 2D frames. This combination of models and independent lifting of features leads to multi-view inconsistent features, compared to our end-to-end learnable semantic and instance features, where we explicitly focus on multi-view and cross-task consistency. In Tab. 10, we compare both semantic and instance segmentation with IGGT. Our approach significantly outperforms IGGT for semantic segmentation on ScanNet (Dai et al., 2017) while being competitive on ScanNet++ (Yeshwanth et al., 2023), as we directly predict semantics with multi-view consistent features rather than lifting separately extracted OpenSeg (Ghiasi et al., 2022) features via VGGT (Wang et al., 2025a) geometry. For instance segmentation, IGGT (Li et al., 2025) achieves competitive results, likely due to the similarity in the underlying contrastive instance learning objective. However, our multi-task formulation, which jointly optimizes several complementary semantic tasks, yields synergistic effects and ultimately leads to improved overall performance. However, IGGT introduces a multi-modal fusion block between the geometry and semantic head, which is missing in UNITE. We believe that future work can investigate the synergy effects between an explicit multi-view loss and the multi-modal alignment between geometry and semantics.

# G   Instance clustering sensitivity

In Fig. 6, we demonstrate the effect of tuning the main clustering hyper-parameter in HDBSCAN 'eps'. The *eps* parameter determines the distance threshold below which clusters will not be further split, effectively acting as a floor to prevent the over-fragmentation of dense regions into smaller micro-clusters, directly impacting the final instance segmentation. We tune the *eps* parameter on the ScanNet training set for optimal alignment to indoor scenes. We hypothesize that this might be a reason why Felzenszwalb (Felzenszwalb & Huttenlocher, 2004) outperforms UNITE on ScanNet++.

# H   Out-of-domain generalization: Outdoor scenes

In Fig. 7, we show UNITE predictions on outdoor scenes from Tanks and Temples (Knapitsch et al., 2017). Because UNITE was trained exclusively on indoor scenes, performing inference on outdoor scenes represents a significant domain shift. To alleviate this problem, methods like OpenScene (Peng et al., 2023) train separate checkpoints for indoor and outdoor data. However, in this section, we aim to demonstrate the generalization performance of UNITE without retraining. Overall, we observe that UNITE struggles to generalize from indoor to outdoor environments. However, for small-scale outdoor scenes, UNITE's instance predictions are accurate enough to separate different instances directly. We believe future work can improve this generalization performance by training on both indoor and outdoor scenes.

**Open-Vocabulary Semantics.** UNITE is distilled from the CLIP representation space; however, if a class or concept was never observed during training, it results in poor alignment with textual features. We observe this when testing UNITE on outdoor scenes, where it is unable to identify even large foreground objects. Despite this, we find that the PCA features remain unique for individual objects. Therefore, we reason that UNITE's embedding space is well-structured and that fine-tuning on outdoor scenes will resolve this lack of understanding of typical outdoor objects.

**Instance Semantics.** Similar to open-vocabulary class semantics, UNITE struggles with the domain shift from indoor to outdoor environments for instance understanding. Differences in appearance, as well as scene and object scale, limit UNITE's instance understanding. However, Fig. 7 shows that on small-scale scenes, UNITE can predict reasonable

Table 9: **Replica Evaluation.** We evaluate UNITE on Replica for 3D semantic segmentation. The Replica dataset contains 51 semantic classes.

|  | mIoU | mAcc |
| --- | --- | --- |
| ConceptFusion (Jatavallabhula et al., 2023) | 10.0 | 17.0 |
| ConcpetGraphs (Gu et al., 2024) | 13.0 | 21.0 |
| HOV-SG (Werby et al., 2024) | 14.4 | 21.2 |
| OpenScene (Peng et al., 2023) | 15.9 | 24.6 |
| **Ours** | **17.0** | **26.6** |

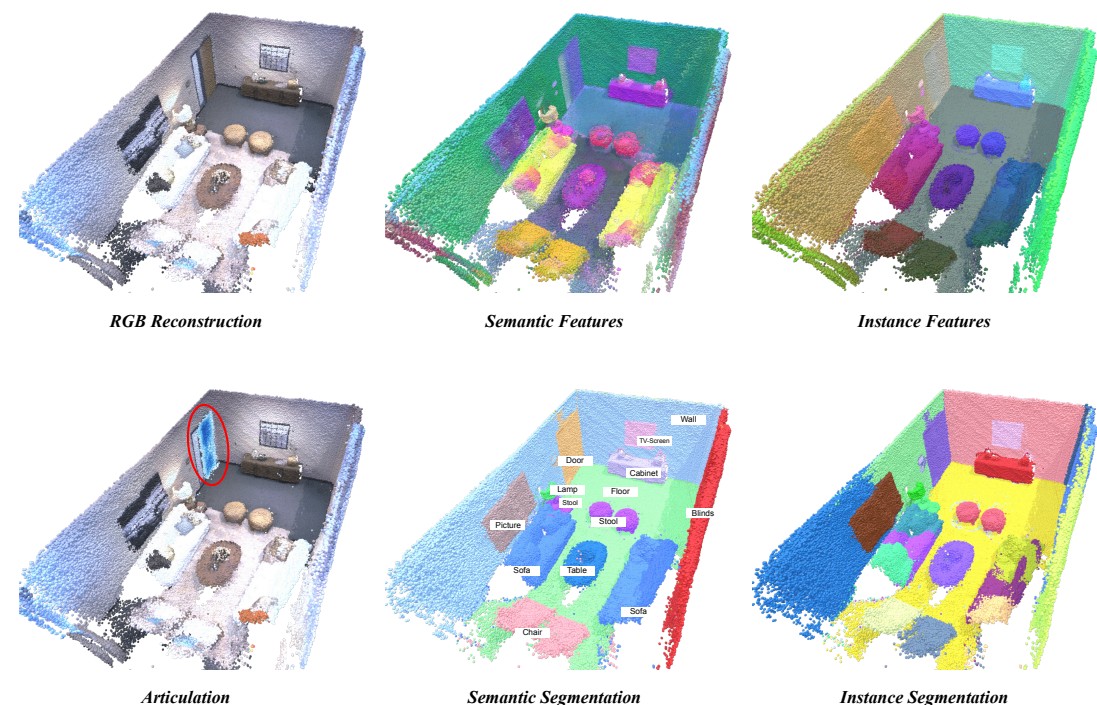

Figure 4: **Results of UNITE on Replica.** UNITE not only produces accurate geometric reconstruction, but also highly distinctive semantic and instance features. These allow for accurate semantic segmentation and instance segmentation results.

instances without being trained on outdoor data. On large-scale scenes with varying object sizes, however, UNITE's instance embeddings struggle to separate objects at a consistently granular level.

# I  Broader Impact Statement

UNITE recovers dense open-vocabulary semantics and articulation directly from RGB images, without requiring pre-built 3D reconstructions or known poses. This lowers the barrier to mapping indoor spaces from casual image collections, which raises privacy concerns. Open-vocabulary querying enables searching reconstructed scenes for arbitrary objects or concepts, including personal or sensitive items, while articulation prediction exposes which parts of a scene are interactive (drawers, cabinets, doors). Together these could facilitate surveillance or unauthorized inference about private environments and their occupants. We encourage deployment to respect consent for captured spaces and to restrict scene querying to authorized users.

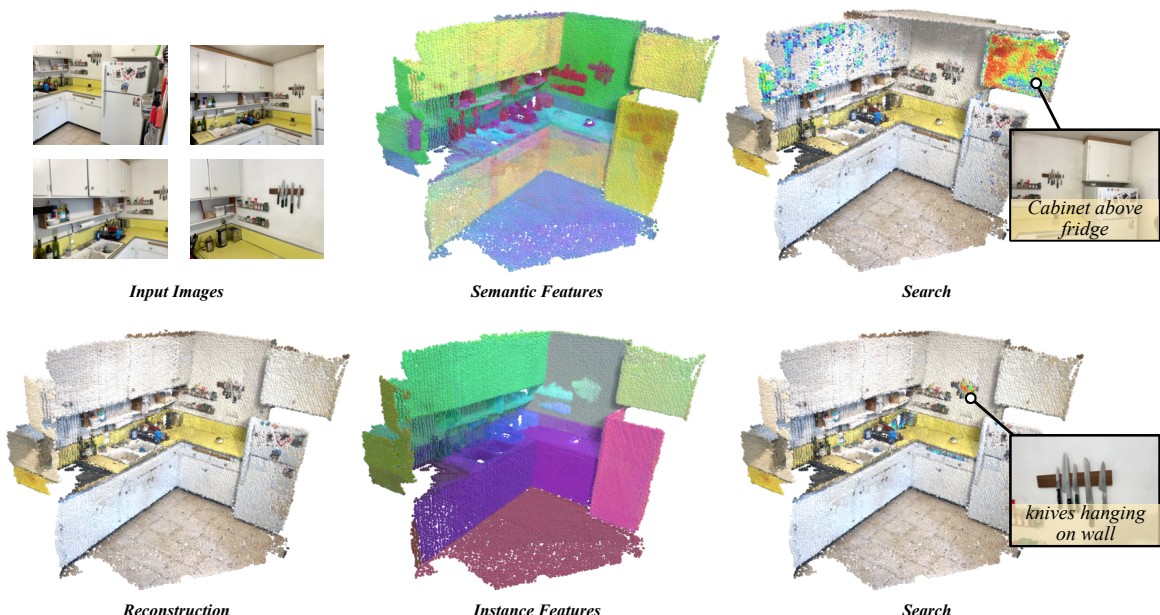

Figure 5: **Results of UNITE on *in-the-wild* scene.** UNITE is able to produce high-quality geometry, instance, and semantic features for *in-the-wild* scenes. The predicted features allow for text-based queries of high complexity, correctly identifying objects from relationship descriptions.

Table 10: **Comparison with IGGT.** 3D semantic segmentation and class-agnostic 3D instance segmentation results. We report only metrics for which IGGT provides values.

| | Semantic Segmentation | | Instance Segmentation | | |
| | ScanNet | ScanNet++ | | ScanNet | |
| | mIoU | mIoU | AP | $AP_{50}$ | $AP_{25}$ |
|---|---|---|---|---|---|
| IGGT (Li et al., 2025) | 39.7 | 20.8 | 12.3 | 24.9 | 41.6 |
| **Ours** | **48.7** | 17.2 | **13.2** | **29.6** | **57.2** |

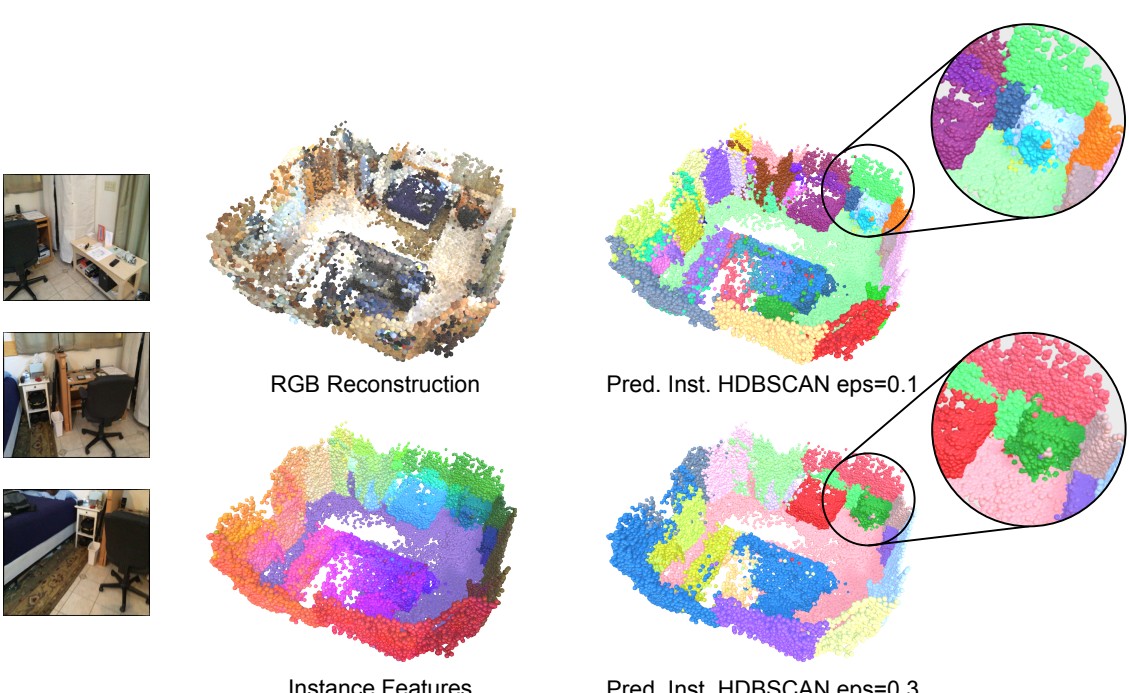

Figure 6: **UNITE sensitivity to hyper-parameters.** UNITE is highly sensitive to the 'eps' hyper-parameter in HDBSCAN used for instance segmentation. Setting *eps* to a small value results in over-segmentation, while a too large value results in under-segmentation. The *eps* parameter is empirically chosen at $eps = 0.1$, which produces balanced results.

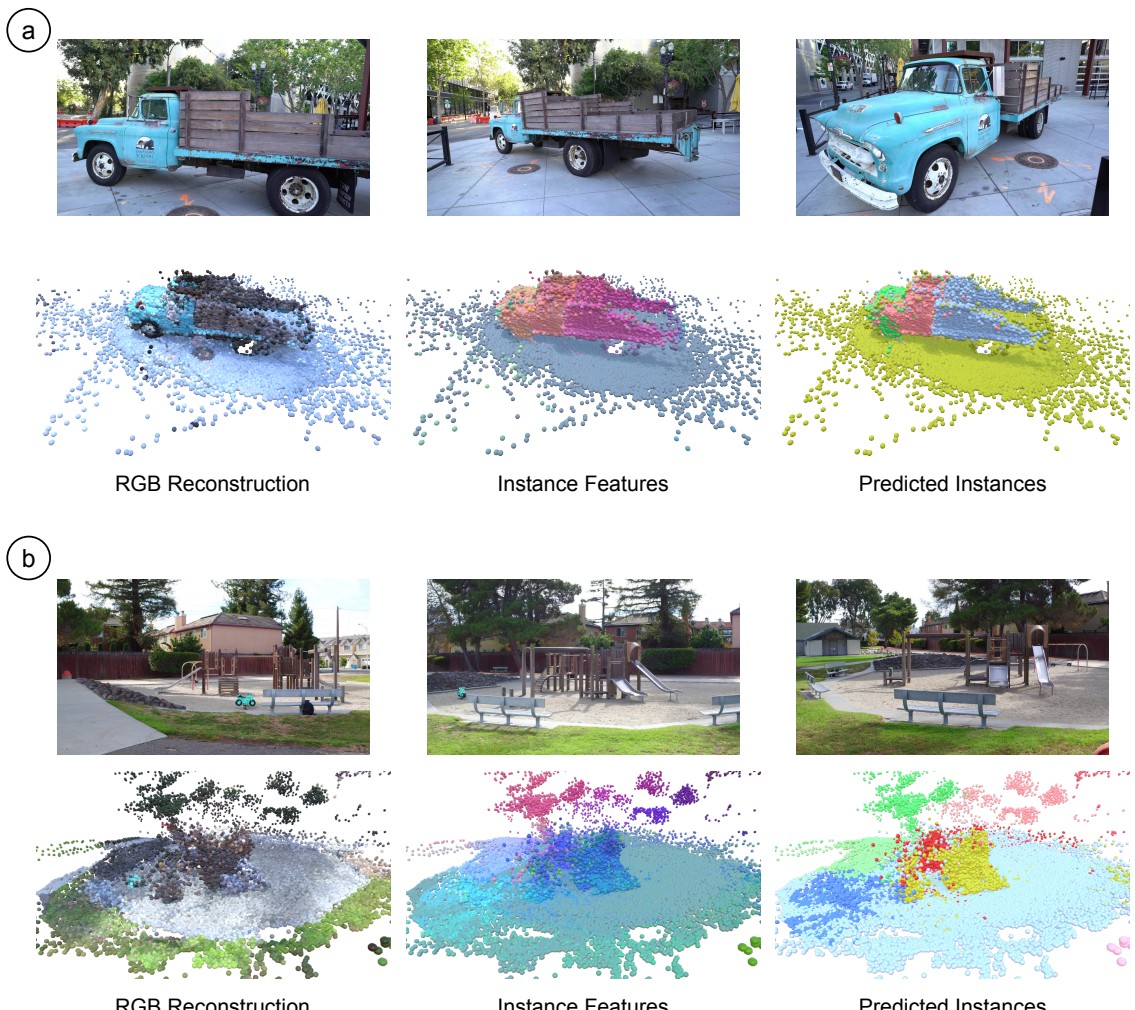

Figure 7: **Outdoor generalization.** UNITE trained on indoor scenes generalizes to outdoor scenes at small scales ⓐ. At larger scales, domain shift in appearance as well as scene and object scales results in poor instance predictions ⓑ.

