# OpenReview forum: "Unified Semantic Transformer for 3D Scene Understanding"
_TMLR — Accepted by TMLR_

### Review · Reviewer_5Z8s · 2026-03-18

**Summary Of Contributions:**

This paper presents UNITE, a Unified Semantic Transformer for holistic 3D scene understanding that reconstructs 3D geometry and predicts dense semantic attributes from RGB images in a single feed-forward pass.

Strength: The paper provides compelling discussion of geometric-semantic mutual enhancement and design choices, and introduces an innovative feature-level multi-view consistency mechanism that effectively bridges 2D foundation model features with 3D geometry.

Weakness: Key limitations include the absence of outdoor scene evaluation for generalization claims and missing geometric reconstruction metrics despite fine-tuning the backbone.

**Audience:**

Yes

**Audience Explanation:**

The paper addresses the timely problem of extending geometric foundation models to semantic understanding, a topic of significant interest to the 3D vision and robotics communities. The multi-view consistency formulation and demonstration of task synergies provide methodological insights relevant to researchers working on 3D representation learning. The feed-forward inference capability offers practical advantages over optimization-based methods for real-time applications.

**Broader Impact Concerns:**

This work positively advances the field by establishing a viable paradigm for unified 3D scene understanding. By demonstrating that geometric reconstruction and semantic understanding can be jointly optimized with mutual benefits, UNITE lowers the barrier for developing holistic 3D perception systems—potentially accelerating progress in embodied AI and robotics where real-time, feed-forward inference is critical. The reliance on self-supervised distillation from foundation models (rather than costly 3D annotations) democratizes access to high-quality 3D semantic reconstruction, enabling broader research participation and applications in resource-constrained settings.

**Claims And Evidence:**

Yes

**Claims Explanation:**

Core claims regarding multi-view consistency and task synergies are convincingly supported by systematic ablations; The experimental design is rigorous.

However, critical evidence is missing for generalization claims (no outdoor evaluation), architectural choices (omitted comparison with SIU3R), and geometric integrity (no post-training reconstruction metrics despite fine-tuning).

**Requested Changes:**

1. Outdoor Scene Evaluation: Please include quantitative or qualitative results on at least one outdoor dataset to validate claims generalization. Since your method trains without semantic annotations, outdoor evaluation should be straightforward and would significantly strengthen the paper's impact.
2. Discussion of SIU3R: Please add a discussion comparing your approach with SIU3R (arXiv:2507.02705) in the Related Work section. Specifically, clarify the trade-offs between your DPT-based feature prediction approach versus their Mask2Former-based semantic map approach. Why is predicting features (and requiring post-hoc clustering) preferable to direct semantic segmentation in your unified framework?
3. Detailed Evaluation Protocol: In the supplementary materials, please provide the exact details of the point cloud preprocessing and nearest-neighbor matching. Since feed-forward methods produce point clouds with different densities and distributions than ScanNet's reconstructed point clouds (used by baseline methods), the preprocessing and matching procedures need explicit documentation to ensure fair comparison with 3D point cloud methods. If possible, provide a set of metrics at the frame-level for fair comparison.
4. Geometric Reconstruction Metrics: Please report geometric performance metrics for the final trained model compared to the VGGT baseline. This is essential to verify that joint semantic training does not degrade geometric capabilities and to substantiate claims about geometric-semantic synergy.

---

> ### Author Response · Authors · 2026-03-27
>
> We thank the reviewer for the thorough and constructive feedback and for recognizing our innovative feature-level multi-view consistency mechanism that effectively bridges 2D foundation model features with 3D geometry. We have revised the manuscript to directly address the requested clarifications regarding outdoor experiments, related work citations and evaluation protocol.
> All changes are highlighted in **blue** in the revised manuscript.
>
>
> *“Please include quantitative or qualitative results on at least one outdoor dataset to validate claims generalization.”*
>
> **Response:**
> We thank the reviewer for this valuable suggestion. In response, we have included qualitative results on two outdoor scenes from Tanks and Temples in the newly added Section F of the appendix (see Figure 7).
> We would like to emphasize that we performed this inference directly, without any retraining or fine-tuning on outdoor data. The results demonstrate promising generalization performance; specifically, on small-scale outdoor scenes, the model's instance predictions are strong enough to separate different objects directly.
> However, we also explicitly discuss the current limitations in the revised text. On large-scale scenes with varying object sizes, the domain shift becomes more apparent, and the instance embeddings struggle to separate objects at a consistently granular level. We believe Figure 7 and the accompanying discussion provide a transparent and balanced view of the model's out-of-domain capabilities.
>
> ---
>
> *“Please add a discussion comparing your approach with SIU3R (arXiv:2507.02705) in the Related Work section.”*
>
> **Response:**
> Thank you for pointing us to SIU3R, we have included it in our related work section. In short, we believe our DPT-based dense feature prediction yields advantages for scaling to a large number of input views (e.g., up to 200 frames) and maintains a continuous open-vocabulary space, whereas Mask2Former-based approaches like SIU3R or PanSt3R are bottlenecked by a fixed number of object queries and struggle to scale efficiently beyond sparse view setups.
>
> ---
>
> *“In the supplementary materials, please provide the exact details of the point cloud preprocessing and nearest-neighbor matching”
>
> **Response:**
> Thank you for raising this point. Reproducibility is important to us, we have added more information for the nearest-neighbor matching in the supplementary and clarified that there is no point cloud preprocessing necessary.
>
> ---
>
> *“Please report geometric performance metrics for the final trained model compared to the VGGT baseline.”*
>
> **Response:**
> Our focus in this paper is on the semantic understanding of feed-forward methods, therefore we keep the geometry prediction heads frozen for our main experiments. However, please find in the appendix Tab. 7 a small ablation on synergy effects between geometry and semantics training at a small scale.
>
> ---
>
> *“If possible, provide a set of metrics at the frame-level for fair comparison.”*
>
> **Response:** In this work, we aim to investigate the impact of semantic feed-forward for 3D scene understanding and therefore focus on point cloud metrics. Nevertheless, we include Tab. 4 for frame level metrics on the open-vocabulary LERF benchmark.

---

### Review · Reviewer_dEqZ · 2026-03-19

**Summary Of Contributions:**

The paper introduces UNITE, a unified multi-task feed-forward transformer for holistic 3D scene understanding from multi-view RGB images. Building on VGGT (Wang et al. 2025) as a geometric backbone, UNITE appends task-specific Dense Prediction Transformer (DPT) decoder heads that jointly predict four dense semantic properties... open-vocabulary semantic features aligned with CLIP, class-agnostic instance embeddings, object articulation parameters, and open-vocabulary search capability. All supervision is derived without manual annotation from SAM and OpenSeg/CLIP, with ground-truth annotations used exclusively for the articulation task. The central training objective combines per-task distillation losses with a confidence-weighted multi-view consistency loss that enforces agreement across different views of the same 3D point, enabling view-invariant semantic and instance representations.

The model is evaluated on six datasets and compared against a representative set of baselines spanning SLAM-based, point-cloud-based, and feed-forward methods. UNITE reports state-of-the-art or competitive performance on semantic segmentation, instance segmentation, and articulation prediction. Ablation studies in Table 4 carefully isolate the effects of distillation training, the multi-view consistency loss, confidence-weighted aggregation, and oracle geometry. Appendix C further examines whether semantic training provides complementary benefit to geometric tasks.

The paper's genuine strengths lie in the breadth of the unified task set, particularly the inclusion of articulation prediction alongside semantic, instance, and open-vocabulary tasks in a single model. The multi-dataset evaluation is thorough and the ablation design is methodologically sound.

Key weaknesses are discussed in the following sections.

**Audience:**

Yes

**Audience Explanation:**

The 3D scene understanding community represents a growing and active segment of TMLR's audience, which increasingly covers applied deep learning and vision research alongside more theoretical Machine Learning. The practical problem that UNITE addresses: holistic, multi-task 3D scene parsing from passive RGB capture is directly relevant to researchers in embodied AI, robotics, spatial computing, and AR/VR, all of whom are represented in TMLR's readership.

The paper's finding that a single end-to-end model can match or exceed task-specific baselines on four different 3D semantic objectives, and that multi-task training produces synergistic rather than competing effects, is a concrete and informative empirical result. The articulation prediction component in particular, predicting object motion directions from multi-view RGB without any manual annotation, is a capability not previously demonstrated within a unified feed-forward architecture, and practitioners building manipulation or interaction systems would find it directly useful.

Even accounting for the overclaimed novelty, the systems level contribution of integrating these tasks coherently, the thorough multi-dataset benchmarking, and the careful ablation design make this a paper whose findings would inform research decisions in the community. TMLR's philosophy of prioritizing technical correctness over novelty thresholds is well-suited to this work... the claims need revision, but the findings themselves are of interest.

**Broader Impact Concerns:**

__The paper does not include a Broader Impact statement.__

The direct applications of holistic 3D scene understanding in robotics, AR/VR, and embodied AI are broadly beneficial. However, a brief statement should acknowledge potential misuse scenarios. The open-vocabulary querying capability - locating specific objects in a 3D reconstruction via free-form text - combined with the model's ability to operate from unposed, casual RGB captures, creates a pipeline with clear dual-use potential in passive spatial surveillance and privacy-invasive scene mapping. The articulation prediction component, which identifies movable parts and their motion directions, may similarly be misused in contexts such as automated vulnerability assessment of physical spaces.

These concerns are modest relative to the state of the technology and are widely shared across the 3D scene understanding literature. The authors do not need to treat them as showstoppers, but a single paragraph Broader Impact section that acknowledges these scenarios and notes any relevant mitigation considerations (e.g. requiring physical camera access to scenes, limitations in real-time performance) would satisfy this requirement.

**Claims And Evidence:**

Yes

**Claims Explanation:**

The core empirical claims are generally well supported. The quantitative results in Tables 1 through 4 are presented under fair experimental conditions - baseline comparisons use public implementations, evaluation is performed consistently in 3D rather than at the view level, and privileged baselines (those requiring depth or ground-truth geometry) are clearly flagged. The ablation study in Table 4 provides a clean incremental analysis that isolates contributions of individual components. The geometry-semantics synergy results in Table 6 are modest and honestly reported, with the authors appropriately noting that margins are small at the current training scale.

However, two categories of claims are not adequately matched by evidence, and their revision is critical.

First, the claim to be the first unified, end-to-end model for multiple 3D semantic tasks is factually incorrect as of the submission date. IGGT (Li et al. arXiv:2510.22706, October 2025; accepted at ICLR 2026) independently introduced a VGGT-based framework with DPT heads and contrastive instance learning two months before UNITE's arXiv preprint. IGGT shares the same backbone (VGGT with DINOv2 tokenization), the same decoder paradigm (DPT heads), the same instance learning strategy (contrastive pull/push loss distilled from SAM family models), and the same feed-forward multi-view semantic reconstruction objective. The paper does acknowledge IGGT as the most similar work, but the comparison is relegated to Appendix E and the fundamental architectural parallelism is not honestly addressed in the main paper. Beyond IGGT, PanSt3R (Zust et al. ICCV 2025, June 2025) demonstrated multi-view consistent feed-forward panoptic segmentation six months prior to UNITE, and Uni3R (Sun et al. August 2025) uses the same VGGT-initialized Cross-View Transformer for joint 3D reconstruction and open-vocabulary semantic understanding. The contribution claims must be revised to reflect this landscape accurately.

Second, the characterization of the multi-view consistency loss as a novel contribution is overstated. Enforcing view-consistent feature agreement with stop-gradient self-distillation draws directly from BYOL and DINO family self-supervised techniques. Confidence-weighted multi-view feature aggregation is a well established technique in depth estimation and visual localization. The specific instantiation here, applying this principle to semantic and instance features within a multi-task 3D reconstruction pipeline, is a reasonable engineering design choice, but the conceptual novelty is limited. The contribution should be framed as an adaptation of established consistency regularization to this setting, not as a new training objective.

A third, narrower concern... the ScanNet++ instance segmentation results in Table 2 show UNITE underperforming the classical Felzenszwalb method on AP and AP50. The provided explanation (Felzenszwalb operates directly on the evaluation mesh, and UNITE performs better on coarser AP25) is plausible but brief. A more complete analysis of when and why the learned instance approach fails to match a geometry-only baseline on a specific dataset would strengthen the paper's honesty about limitations.

**Requested Changes:**

__Critical__

1 - Revise all contribution claims and the related work section to accurately position UNITE within the concurrent literature. The claim of being the first unified end-to-end model must be removed or precisely scoped. The comparison with IGGT must be moved into the main paper with an explicit discussion of the architectural similarities and the specific ways UNITE extends or differs from it. Concretely, the authors should address - why the cross-modal fusion block in IGGT is not used... how the confidence-weighted consistency loss compares to IGGT's multi-view instance learning design; and what quantitative gains are attributable to the additions UNITE makes over the IGGT design pattern. The Appendix E comparison is a good starting point but is too brief to satisfy this requirement.

2 - Reframe the multi-view consistency loss contribution. Rather than presenting Equations 3 through 6 as a novel training objective, the paper should characterize them as an application of known self-supervised consistency regularization (citing the relevant prior literature including BYOL, DINO, and multi-view consistency work in depth estimation) to the specific problem of multi-task semantic 3D reconstruction. The contribution is the specific application and the evidence that it helps in this context, both of which are legitimate and worth reporting.

3 - Provide a more thorough failure mode analysis for the ScanNet++ instance segmentation results. The paper currently reports that UNITE underperforms Felzenszwalb on AP and AP50 on this dataset but offers only a brief attribution to evaluation mesh access. Qualitative examples showing where the learned instance features fail (e.g., fine grained object boundaries, densely cluttered scenes) would make the paper more informative and honest about the approach's current limitations.

4 - Clarify the articulation prediction results in Table 3 more carefully. The multi-task UNITE model outperforms the task-only variant on most metrics, but the recall values for both motion type and movable part detection are low across all methods. The paper should discuss whether this reflects a fundamental limitation of the regression formulation, the linearization of rotational motion, or the small size of the MultiScan training set relative to model capacity.



__Strengthening (Not Critical)__

1 - An ablation directly comparing UNITE's confidence weighted consistency loss against IGGT's Cross-Modal Fusion Block would clarify whether the two approaches offer complementary benefits and help readers understand the design tradeoffs.

2 - The multi-task loss weighting is presented without a sensitivity analysis. Task weighting is a known source of variance in multi-task learning; some evidence of robustness to perturbations of these weights, even if brief, would add confidence in the reported results.

3 - The linearization of rotational motion (approximating rotation by the displacement of surface points rotated 90 degrees) is an unusual design choice that the paper justifies only briefly. A qualitative or quantitative comparison with direct axis-angle regression or a normalized direction vector approach would help readers assess whether this approximation introduces systematic errors for objects with small angular range of motion.

4 - An extended failure case and out-of-distribution analysis beyond the Replica and LERF scenes in the appendix would strengthen the practical value of the paper. In particular, outdoor scenes or scenes with significant dynamic content would help characterize generalization limits.

---

> ### Author Response · Authors · 2026-03-27
>
> We thank the reviewer for recognizing our thorough evaluation, methodologically sound ablations, and fair experimental conditions (Tabs. 1-4). We revised the manuscript (changes in **blue**) to address your requested clarifications on claim calibration, oracle geometry, and main/appendix consistencies.
>
> *“First, the claim to be the first unified, end-to-end model for multiple 3D semantic tasks is factually incorrect as of the submission date”*
>
> **Response:**
>
> We agree IGGT is highly relevant and predates our work, which we acknowledge in related work. We do not believe we claimed to be the first; if we are wrong, please point us to the sentence so we can resolve it.
>
> Uni3R, PanSt3R, and IGGT primarily focus on single tasks and lack full end-to-end pipelines. Uni3R merges semantic Gaussians with separate LSeg features; PanSt3R requires custom post-processing; and while IGGT predicts instances end-to-end, its semantic segmentation uses separate 2D OpenSeg features. Ultimately, only UNITE predicts semantic, instance, and articulation features in a single feed-forward pass.
>
> ---
>
> *“The comparison with IGGT must be moved into the main paper with an explicit discussion of the architectural similarities and the specific ways UNITE extends or differs from it.”*
>
> **Response:**
>
> We will move IGGT's results into Tabs. 1 and 2. As acknowledged in our related work and results, both methods share a similar instance learning approach. Both are inspired by bottom-up instance seg. methods like SGPN, OccuSeg, and GARField, which predict instance embeddings before regressing them into instances. Ultimately, our focuses are different: IGGT prioritizes cross-modal alignment, whereas UNITE emphasizes explicit multi-view consistent features. We believe these concepts are highly complementary, combining them offers a natural path for future work, which we now highlight in our appendix discussion on IGGT.
>
> ---
>
> *“Second, the characterization of the multi-view consistency loss as a novel contribution is overstated.“*
>
> **Response:**
> Thank you for pointing us to BYOL and DINO, we added citations in the main paper. We agree that there are certain implementation similarities, yet none of these representation learning approaches investigates multi-view consistent feature learning, therefore we believe that our proposed multi-view consistency loss remains novel even if implementation similarities to representation learning approaches and confidence-based loss regularizations exist (cited now as well). To the best of our knowledge no similar approach exists that exploits predicted confidences for multi-view semantic consistency.
>
> ---
>
> *“Clarify the articulation prediction results in Table 3 more carefully [...], recall values [...] are low across all methods”*
>
> **Response:** Thank you for raising this point. We attribute the lower recall, as a precision-recall trade-off. In Eq. 9 we introduce a focal loss to handle severe class imbalance between static background and articulated pixels. Optimizing the focal loss hyper-parameters is inherently difficult and has direct impact on precision-recall, nevertheless our reported F1 score shows our configuration balances precision and recall effectively. We added a discussion of Tab. 3's results in the context of this trade-off to the paper.
>
> ---
>
> *“The multi-task loss weighting is presented without a sensitivity analysis.”*
>
> **Response:** We agree loss weighting has a big impact on multi-task learning. We use naive equal weighting (reported in the appendix.) because our goal is to demonstrate the potential of unified feed-forward models for image-only 3D scene understanding. We believe, our loss weighting and other design choices are not upper bounds, leaving room for future work.
>
> ---
>
> *“The linearization of rotational motion (approximating rotation by the displacement of surface points rotated 90 degrees) is an unusual design choice that the paper justifies only briefly.”*
>
> **Response:** We linearize rotational motion over an axis representation to phrase it as a dense task, utilizing the unified design of dense DPT heads for prediction. We updated the method section to clarify this motivation.
>
> ---
>
> *“A more complete analysis of when and why the learned instance approach fails to match a geometry-only baseline on a specific dataset would strengthen the paper's honesty about limitations.”*
>
> **Response:** Thank you for raising this point, together with a request from another review we added Fig. 6 and Fig. 7 in the appendix which investigate failure cases and hyper-parameter sensitivity for instance segmentation.
>
> ---
>
> *“An extended failure case and out-of-distribution analysis beyond the Replica and LERF scenes in the appendix would strengthen the practical value of the paper.”*
>
> **Response:** We thank the reviewer for recognizing our OOD analysis in the appendix. We added outdoor qualitative experiments in Fig. 7 demonstrating a bigger domain generalization but also failure cases of our work.

---

> > ### Comment · Reviewer_dEqZ · 2026-04-01
> > **Response to Authors' Comments**
> >
> > __"We did not claim to be first"__
> > - This precise argument does not currently appear in the main paper. The abstract and contribution bullets create an implied primacy without stating the specific technical basis for it. The revision should make this argument explicitly in the abstract and in the related work comparison with IGGT, rather than leaving readers to infer it.
> >
> > __"Moving IGGT into the main paper"__
> > - Appreciate that IGGT results will be added to Tables 1 and 2. The characterization of the two methods' different emphases is helpful. However, the main paper's related work section should also state directly that UNITE and IGGT share the same geometric backbone (VGGT with DINOv2), the same DPT decoder architecture, and the same contrastive instance learning paradigm trained from SAM masks. The differences UNITE introduces -- the multi-view semantic consistency loss, the articulation head, and the elimination of frozen 2D modules -- are meaningful and worth stating, but they should be articulated against an honest account of the shared foundation rather than presented in isolation.
> >
> > __"The multi-view consistency loss"__
> > - The response is accepted. The argument that no prior work has applied confidence-weighted multi-view semantic consistency within a multi-task 3D reconstruction pipeline is reasonable, and the addition of BYOL and DINO citations satisfies the framing requirement. No further revision needed on this point.
> >
> > __"Articulation recall and focal loss"__
> > - The explanation is technically sound and the addition of a trade-off discussion to the paper satisfies this concern. No further revision needed.
> >
> > __"Multi-task loss weighting"__
> > - Accepted. For a systems demonstration paper, equal weighting as a proof-of-concept baseline is a reasonable and honest choice, provided it is reported as such rather than as an optimized design. This is already done in the appendix. No further revision needed.
> >
> > __"Linearization of rotational motion"__
> > - The architectural consistency motivation is now clear and the updated method section satisfies this concern. No further revision needed.
> >
> > __"Failure cases"__
> > - The addition of figures, along with outdoor OOD results, is a meaningful improvement. No further revision needed.

---

> > > ### Author Response · Authors · 2026-04-04
> > >
> > > Thank you for engaging with our responses and acknowledging that our revisions have strengthened the paper.
> > >
> > > Following your remaining suggestions, we have added a sentence in the introduction that explicitly recognizes prior semantic feed-forward models by citing them and highlights IGGT separately as the most closely related work. We have also added a sentence in the related work section to clearly emphasize both the similarities and differences of our approach with respect to IGGT.

---

### Review · Reviewer_oQKR · 2026-03-24

**Summary Of Contributions:**

This paper introduces UNITE, a unified feed-forward transformer for multiple 3D scene understanding tasks from multi-view RGB images, including semantic segmentation, instance segmentation, open-vocabulary search, and object articulation. The central idea is to build a shared multi-view geometric backbone with task-specific semantic heads, trained through 2D foundation-model distillation together with a multi-view consistency loss to encourage 3D-coherent predictions across views.

The paper’s main strengths are that the overall direction is timely and important, the unified treatment of several dense 3D semantic tasks is ambitious, and the multi-view consistency design is conceptually well motivated. The empirical results also suggest meaningful multi-task synergy, especially for articulation-related prediction.

The main weaknesses are that some of the paper’s central claims appear stronger than what is directly supported by the experiments, particularly regarding how fully end-to-end the system is in practice. In addition, the main semantic results seem closely tied to oracle geometry, but this dependency is not sufficiently transparent in the main text. There also appear to be inconsistencies between the main paper and the appendix regarding methodological details, which hurt clarity and reproducibility.

**Audience:**

Yes

**Audience Explanation:**

The paper addresses a problem that is clearly relevant to parts of the TMLR audience, especially researchers working on 3D scene understanding, multi-view learning, and unified architectures for dense semantic prediction. The goal of handling multiple 3D understanding tasks within a shared feed-forward framework is timely and potentially impactful. Even though I have concerns about whether all of the paper’s claims are fully supported in the current version, I believe the problem setting, methodology, and empirical findings would still be of interest to at least some readers in the TMLR community.

**Broader Impact Concerns:**

I do not have specific broader impact concerns that require additional discussion beyond standard considerations for 3D scene understanding and visual perception systems. My main concerns are about claim-evidence alignment, clarity, and reproducibility rather than ethical or societal impact.

**Claims And Evidence:**

No

**Claims Explanation:**

The paper presents an interesting and potentially impactful direction, but I do not think its main claims are currently supported by sufficiently clear and convincing evidence. In particular, the relationship between the main semantic results and the oracle-geometry setting is not made sufficiently transparent, which makes it difficult to assess how end-to-end the method actually is in practice. In addition, there appear to be inconsistencies between the main text and the appendix regarding important methodological details, which hurt clarity and reproducibility. More broadly, some of the paper’s headline claims seem stronger than what is directly established by the experiments. For these reasons, I do not think the current version yet provides fully accurate, convincing, and clear support for all of its central claims.

**Requested Changes:**

Critical to securing my recommendation for acceptance:

1. Clarify the role of oracle geometry in the main results.
The paper should explicitly state in the abstract, main text, and table captions whether the main semantic results rely on ground-truth depth and pose. As currently presented, the relationship between the main results and the oracle-geometry setting is not sufficiently transparent. Because this point directly affects how readers interpret the paper’s central claims, the revision should also include a clear comparison between performance with predicted geometry and performance with oracle geometry in the main text.

2. Resolve inconsistencies between the main text and the appendix.
The descriptions of the method should be fully consistent across all parts of the paper. In particular, the paper should reconcile the discrepancy regarding the semantic teacher, as well as the discrepancy regarding the articulation representation. These inconsistencies currently make it difficult to determine the exact method and training setup, and they significantly weaken reproducibility.

3. Calibrate the paper’s claims to the evidence.
Several of the paper’s headline claims should be revised so that they more precisely reflect the actual experimental setting and evidence. In particular, statements about being “fully end-to-end,” “mostly self-supervised,” and highly efficient should be toned down or clarified where the method depends on oracle geometry, explicit supervision, or nontrivial runtime. This revision is important because the current presentation overstates what is directly established by the experiments.

Would strengthen the work:

1. Move key quantitative evidence from the appendix to the main paper.
Since open-vocabulary capability and efficiency are presented as important contributions, the corresponding quantitative results should appear in the main text rather than only in the appendix. This would make the paper easier to evaluate and would better align the presentation with the paper’s stated contributions.

2. Improve analysis of limitations and sensitivity.
The paper would be stronger with a brief discussion of limitations and failure cases, especially in settings where competing approaches perform better. If applicable, a short sensitivity analysis for important design choices or hyperparameters would also help readers better understand the robustness of the method.

---

> ### Author Response · Authors · 2026-03-27
>
> We thank the reviewer for the thorough and constructive feedback and for recognizing the timeliness and impactfulness of our novel unified feed-forward scene understanding approach. We have revised the manuscript to directly address the requested clarifications regarding calibration of claims, oracle geometry in evaluations, and main/appendix consistencies.
> All changes are highlighted in **blue** in the revised manuscript.
>
> --------------
>
> *“Clarify the role of oracle geometry in the main results. The paper should explicitly state in the abstract, main text, and table captions whether the main semantic results rely on ground-truth depth and pose”*
>
> **Response:**
> We agree that the use of oracle geometry in the paper could have been introduced more clearly. We employ oracle geometry only to make a comparable and fair comparison with the segmentation methods in Tabs. 1-2. UNITE does not require oracle geometry and can be run exclusively with RGB frames.  Tab. 4 shows that the performance difference w/ and w/o oracle geometry is marginal. We updated the table captions to make this explicit as well as highlighted this again in 4.1 Setup.
>
> --------------
>
> *“Resolve inconsistencies between the main text and the appendix. The descriptions of the method should be fully consistent across all parts of the paper. “*
>
> **Response:**
> We thank the reviewer for pointing this out. We provide minor corrections regarding the use of OpenSeg and CLIP in the main paper, to make clear that we use OpenSeg which encodes in CLIP feature space.
> Regarding the articulation, we made a minor change in the main paper to make it more correctly aligned with the appendix description which provides direct implementation details.
>
> --------------
>
> *“Statements about being “fully end-to-end,” “mostly self-supervised,” and highly efficient should be toned down or clarified where the method depends on oracle geometry, explicit supervision, or nontrivial runtime.”*
>
> **Response:**
> Thank you for raising this point, our goal is to support our claims with experimental evidence. We hope that we have already clarified that our approach does not need any oracle geometry at inference time. The tasks ‘semantic segmentation’, ‘instance segmentation’ and ‘open-vocabulary reasoning’ are learned self-supervised, only for articulation prediction, we rely on annotations.
> Regarding highly efficient, we believe Tab. 6 supports the efficiency characteristics of our method. Compared to an ensemble of experts using lifting for instance employing VGGT+CLIP (Tab. 4)  we see large improvements in speed ~34s vs. ~5min / scene.
> Finally, regarding fully end-to-end: UNITE predicts fully end-to-end dense semantic, instance and articulation features that are directly usable for 3D scene understanding tasks such as 3D semantic segmentation, instance segmentation and articulation prediction. Some post-processing is required to align the predictions with ground truth labels, however this is in contrast to methods such as IGGT which combine learned end-to-end instance segmentation with semantic features separately extracted using OpenSeg.
> Overall, we agree that the claims in the Abstract and Introduction are very strong and we introduced minor corrections to make sure that our claims are supported by our provided evidence.
>
> ------
>
> *“Since open-vocabulary capability and efficiency are presented as important contributions, the corresponding quantitative results should appear in the main text rather than only in the appendix. “*
>
> **Response:**
> Thank you for this pointer, we moved the quantitative LERF results into the main paper.
>
> -----
>
> *“Improve analysis of limitations and sensitivity. The paper would be stronger with a brief discussion of limitations and failure cases, especially in settings where competing approaches perform better.”*
>
> **Response:**
> Aligned with comments from other reviewers we introduced Fig. 6 and Fig. 7 in the appendix which investigates failure cases and hyper-parameter sensitivity for instance segmentation.

---

### Review · Reviewer_c2ku · 2026-04-08

**Summary Of Contributions:**

The paper is essentially trying to do full 3D scene understanding from multi-view RGB in one model, instead of the usual setup where geometry, semantics, etc. are handled separately.

The proposed model jointly predicts geometry, semantic features, instance structure, open-vocabulary representations, and articulation, all within a single feed-forward transformer. A key piece is the multi-view consistency loss, which enforces that features corresponding to the same 3D point agree across views.

I think the direction makes a lot of sense. Unifying these tasks is hard, and the way they build on a shared multi-view backbone is good. The results are also reasonable across several tasks, which suggests that the multi-task setup is actually working.

My main reservation is around how the method is described. It still relies quite a bit on distillation from 2D models and mixed supervision, so calling it fully end-to-end or mostly self-supervised feels a bit overstated.

**Audience:**

Yes

**Audience Explanation:**

The problem addressed is clearly relevant to the TMLR audience, especially in 3D vision and multimodal learning. The model itself could also be useful in adjacent areas like AR/VR and 3D graphics and various fields in Robotics, Embodied AI etc.

More broadly, the idea of replacing task-specific pipelines with a single architecture that jointly models geometry and semantics is aligned with where the field is going. Even beyond the specific results, this direction is important and likely to attract interest.

**Broader Impact Concerns:**

I do not see any significant broader impact concerns beyond standard considerations for 3D perception systems. The work primarily advances technical capabilities, and potential risks are not unique to this paper.

**Claims And Evidence:**

Yes

**Claims Explanation:**

The experimental results are generally solid and align with what the paper is trying to claim. The model performs well across multiple tasks, and the idea of learning a shared representation with multi-view consistency seems to be working in practice.

At the same time, some of the higher-level claims are a bit stronger than what is strictly demonstrated. In particular, the method depends on distillation from 2D foundation models (e.g., segmentation and vision-language features), and also uses explicit supervision for certain components like articulation. So while the system is trained jointly and behaves like a unified model at inference, the learning setup itself is not entirely self-supervised or purely end-to-end.

I don’t think this takes away from the core contribution, but it would help if the paper was more precise in how these aspects are described. As it stands, the evidence supports the main idea, but some of the wording could be better aligned with the actual setup.

**Requested Changes:**

1. The "end-to-end" and "self-supervised" framing is directionally correct, but still a bit stronger than necessary. The method is unified at inference, but training relies on multiple supervision signals. Slightly softening the wording would improve precision without weakening the contribution.

2. The ablation in Table 5 is helpful, but the gains from multi-view consistency and other methods are noticeably smaller than those from distillation alone. A more explicit discussion of this gap would help clarify what is actually driving performance.

---

> ### Author Response · Authors · 2026-04-09
>
> We thank the reviewer for the thorough and constructive feedback and for recognizing our idea of learning a shared representation with multi-view consistency as well as our evaluation efforts. In line with the feedback from other reviewers, we have revised the manuscript to directly address the requested wording changes in ***blue*** and provide context for Tab. 5 here.
>
> We agree that when comparing with a mixture-of-experts baseline (VGGT+CLIP), distillation into a natively consistent 3D scene understanding model yields the highest gains; this is part of our core contributions. However, we observe that naive distillation from 2D features contains view-specific inconsistencies that produce contradictory training signals for the same 3D point. Our multi-view consistency loss with confidence weighting directly addresses this, yielding an additional +2.8 mAcc (4.5% relative improvement, see Tab. 5), a meaningful gain given that it is build on top of an already strong distillation baseline. We note that diminishing marginal returns are expected: the consistency loss crucially refines/stabilizes an already solid feature space rather than introducing new semantic information.

---

### Review · Reviewer_RZRN · 2026-04-27

**Summary Of Contributions:**

This paper presents UNITE, a unified feed-forward transformer for holistic 3D scene understanding from multi-view RGB images. The model builds on a geometric feed-forward backbone vggt and adds dense prediction heads for open-vocabulary semantic features, class-agnostic instance embeddings, and object articulation. The method is trained using 2D foundation-model distillation, task-specific losses, and a confidence-weighted multi-view consistency loss that encourages features corresponding to the same 3D point to agree across views. The paper evaluates the method across several 3D scene understanding tasks and provides includes ablations for distillation, multi-view consistency, confidence weighting, oracle geometry, multi-task training, runtime, geometry/semantics interaction, and out-of-domain behavior.

**Audience:**

Yes

**Audience Explanation:**

The paper addresses an important problem: replacing fragmented 3D perception pipelines with a unified model that jointly predicts geometry-grounded semantic, instance, open-vocabulary, and articulation information purely from image inputs. This is likely to be of interest to researchers in 3D vision, representation learning, robotics, embodied AI, AR/VR, and spatial computing.

**Broader Impact Concerns:**

I do not see major broader impact concerns beyond standard issues for 3D perception systems. However, the paper could briefly acknowledge that open-vocabulary 3D scene querying and articulation prediction may have dual-use implications for privacy-invasive scene mapping or automated analysis of physical spaces. These concerns are not unique to this work and are not a reason to reject the paper, but a short statement would be appropriate.

**Claims And Evidence:**

Yes

**Claims Explanation:**

The core empirical claims are now mostly supported with the revisions (positioning relative to IGGT, outdoor evalution, and narrowed contribution to only indoor settings)
- The revised manuscript provides broad experimental evidence across semantic segmentation, instance segmentation, articulation, and open-vocabulary search.
- The ablation study is also helpful. It shows that distilling 2D foundation-model features into a unified feed-forward model is the main source of improvement over a VGGT+CLIP lifting baseline, while multi-view consistency and confidence weighting provide additional but smaller gains.
- The paper now more clearly acknowledges that UNITE shares important design elements with IGGT and related feed-forward 3D understanding work, while differing in its explicit multi-view semantic consistency, unified semantic/instance/articulation prediction, and lack of frozen 2D modules at inference for open-vocabulary semantics.

**Requested Changes:**

I am generally positive about the revised manuscript, but I would like to raise several additional points that were not fully emphasized in the previous discussion.
- the confidence-weighted multi-view consistency loss needs further clarification. This is particularly important if the confidence is directly inherited from the VGGT geometric backbone. Prior experience in the community suggests that predicted confidence maps do not always faithfully capture where the geometry is reliable, nor how much trust should be assigned to a given view. Since Table 5 attributes a nontrivial improvement to confidence weighting, the paper should clarify the role of this confidence in the multi-view consistency objective, for example through visualizations or ablations.
- I think the paper would benefit from a direct teacher / pseudo-label baseline. The semantic head is distilled from SAM/OpenSeg-style 2D features, and the instance head is trained from SAM masks lifted and merged in 3D. A direct evaluation of these lifted teacher pseudo-labels under the same frame and geometry setting would help determine how much UNITE improves over the teacher pipeline, rather than only compressing or denoising it.
-  the paper reports runtime for different numbers of input frames, but does not report accuracy as a function of the number of input views. Since the model is trained on short image sequences but evaluated with up to 200 frames, an accuracy-vs-views ablation, e.g., 2/5/10/20/50/100/200 frames with random subset variance, would make the claims about arbitrary-view feed-forward inference much more convincing.

---

> ### Author Response · Authors · 2026-05-11
>
> We thank the reviewer for responding to our revisions, which improve the clarity of our paper and positioning to existing related work.
>
> **If the confidence is directly inherited from the VGGT geometric backbone.**
>
> ***Response***: We follow the confidence formulation from VGGT (originally introduced in DUSt3R), but learn separate confidence maps for each semantic head rather than reusing the geometric confidence from VGGT. We agree that predicted confidence is not a perfect indicator of feature reliability. Nonetheless, our ablation study in Tab. 5 shows that applying the confidence formulation yields consistent improvements in feature stability and downstream performance.
>
> **I think the paper would benefit from a direct teacher / pseudo-label baseline. **
>
> ***Response***: We agree that a pseudo-label baseline is important. Indeed this is exactly what we are showing with "VGGT+CLIP Lifting" in Tab. 5: VGGT for the geometry features and CLIP for the teacher semantic features. The CLIP features here are derived from OpenSeg which projects into the same CLIP feature space, which is why we called it VGGT+CLIP. We understand that this causes confusion and will rename it to make the teacher/pseudo-label nature explicit in the final version.
>
> **Report accuracy as a function of the number of input views**
>
> ***Response***: Thank you for raising this point. We see the interest in a reduced number of views for 3D scene understanding, which drives computational efficiency, however a core contribution of UNITE is that it scales up to 200 frames to support scene-level inference, since 5 or 10 views are not enough to capture and understand large-scale indoor scenes. We have therefore provided results for 50/100/150/200 views by sampling frames equidistantly per scene on ScanNet for UNITE:
>
> | Views | mIoU | mAcc |
> |---:|---:|---:|
> | 50 | 29.9 | 37.6 |
> | 100 | 38.5 | 52.1 |
> | 150 | 47.4 | 67.6 |
> | 200 | 48.7 | 68.3 |
>
> We see that there is a big performance gap between 50 and 200 views since scene coverage is severely limited with fewer frames, but there is only a small difference in performance between 150 and 200 frames, indicating that more frames alone will not improve results much further once view coverage of the scene is good enough. We will include this ablation in the final version.

---

### Decision · Action_Editor_2kyB · 2026-05-10

**Recommendation:** Accept with minor revision

**Additional Comments:**

For the minor revision, please address the following items in the final version:

1. Clarify that the semantic confidence maps are learned separately and are not directly reused from the VGGT geometric confidence. Also briefly state how confidence weighting is used in the multi-view consistency loss.

2. Rename or clarify the "VGGT+CLIP Lifting" baseline as a lifted teacher / pseudo-label baseline based on OpenSeg/CLIP features.

3. Include the reported ScanNet view-number ablation for 50, 100, 150, and 200 views, with a brief note on the saturation from 150 to 200 views.

4. Add a short broader-impact statement on possible privacy risks from open-vocabulary 3D scene querying and articulation prediction.

These are minor final-version clarifications.

**Audience:**

Yes

**Audience Explanation:**

Yes. The paper addresses unified 3D scene understanding from multi-view RGB images, which is of clear interest to researchers in 3D vision, multi-view geometry.  The most useful contribution is a coherent feed-forward system that jointly predicts semantic features, instance embeddings, open-vocabulary features, and articulation information.

Even though the novelty is best understood as system-level task unification rather than a fundamentally new modeling paradigm, the empirical findings and design analysis should be useful to a meaningful part of the TMLR audience.

**Claims And Evidence:**

Yes

**Claims Explanation:**

The claims are sufficiently supported after revision. The paper presents broad experimental evidence across semantic segmentation, instance segmentation, open-vocabulary querying, and articulation prediction, with ablations covering distillation, multi-view consistency, confidence weighting, oracle geometry, multi-task training, runtime, and out-of-domain behavior.

Earlier concerns have been addressed to a degree that meets the TMLR bar. The remaining concerns mainly concern further clarification of confidence weighting and additional useful ablations, rather than blocking evidence gaps.